# Numerical Data Imputation for Multimodal Data Sets: A Probabilistic Nearest-Neighbor Kernel Density Approach

**Lalande Florian** *florian.lalande@oist.jp*
*Neural Computation Unit*
*Okinawa Institute of Science and Technology*
*1919-1 Tancha, Onna-son, Okinawa, JAPAN*

**Doya Kenji** *doya@oist.jp*
*Neural Computation Unit*
*Okinawa Institute of Science and Technology*
*1919-1 Tancha, Onna-son, Okinawa, JAPAN*

**Reviewed on OpenReview:** *https://openreview.net/forum?id=KqR3rgooXb*

## Abstract

Numerical data imputation algorithms replace missing values by estimates to leverage incomplete data sets. Current imputation methods seek to minimize the error between the unobserved ground truth and the imputed values. But this strategy can create artifacts leading to poor imputation in the presence of multimodal or complex distributions. To tackle this problem, we introduce the $k$NN$\times$KDE algorithm: a data imputation method combining nearest neighbor estimation ($k$NN) and density estimation with Gaussian kernels (KDE). We compare our method with previous data imputation methods using artificial and real-world data with different data missing scenarios and various data missing rates, and show that our method can cope with complex original data structure, yields lower data imputation errors, and provides probabilistic estimates with higher likelihood than current methods. We release the code in open-source for the community[1].

## 1 Background and related work

As sensors are now ubiquitous and the Internet of Things has become widespread and found numerous applications, Big Data is often referred to as the "Gold of the 21st Century". However, along with the proliferation of numerical databases, missing data has become a pervasive problem: they can introduce a bias, lead to wrong conclusions, or even prevent from using data analysis tools that require complete data sets.

To mitigate this issue, data imputation algorithms have been developed. From the straightforward mean/mode imputation (Little & Rubin, 2014) to recent generative adversarial networks (GAN) models (Yoon et al., 2018), a wide range of tools are available to impute incomplete data sets. As the variety and specificity of available data imputation algorithms can be overwhelming for practitioners, flexible packages like `DataWig` allow optimal imputation results by sweeping through several methods and automatically perform hyper-parameter tuning (Bießmann et al., 2019).

Data imputation most popular application consists of recovering missing parts of an image, also known as inpainting. Deep learning methods have shown promising results for image inpainting and are therefore the preferred solutions for image recovery (Xiang et al., 2023). However, typical image features differ from tabular data. This study focuses on tabular numerical data sets, that is numerical real-valued data arranged in rows and columns in a form of a matrix. For numerical data sets, recent benchmarks argue that deep-learning imputation methods do not perform better than simple traditional algorithms (Bertsimas et al.,

---

[1]https://github.com/DeltaFloflo/knnxkde

2018; Poulos & Valle, 2018; Jadhav et al., 2019; Woznica & Biecek, 2020; Jäger et al., 2021; Lalande & Doya, 2022; Grinsztajn et al., 2022). These studies show that the $k$NN-Imputer (Troyanskaya et al., 2001) and MissForest (Stekhoven & Bühlmann, 2012), in spite of being simple algorithms, generally perform better over a large range of data sets in various missing data scenarios. In the presence of linear dependencies, Multiple Imputation using Chained Equations (MICE) and its variants (van Buuren & Groothuis-Oudshoorn, 2011; Khan & Hoque, 2020) can show good imputation performances.

We denote $x \in \mathbb{R}^D$ the complete ground truth for an observation in dimension $D \geq 2$, and $m \in \{0, 1\}^D$ the missing mask. The observed data is presented as $\tilde{x} = x \odot m$, where $\odot$ denotes the element wise product. Data may be missing because it was not recorded, the record has been lost, degraded, or data may alternatively be censored. The exercise now consists in retrieving $x$ from $\tilde{x}$, while allowing incomplete data for modeling, and not only complete data.

The probability distribution of the missing mask, $p(m)$ is referred to as the missing data mechanism (or missingness mechanism), and depends on missing data scenarios. Following the usual classification of Little and Rubin, missing data scenarios are split into three types (Little & Rubin, 2014): missing completely at random (MCAR), missing at random (MAR) and missing not at random (MNAR).

In MCAR the missing data mechanism is assumed to be independent of the data set and we can write $p(m|x) = p(m)$. In MAR, the missing data mechanism is assumed to be fully explained by the observed variables, such that $p(m|x) = p(m|\tilde{x})$. The MNAR scenario includes every other possible scenarios, where the reason why data is missing may depend on the missing values themselves.

Numerical data imputation methods are usually evaluated using the normalized RMSE (NRMSE) between the imputed value and the ground truth. The higher the average NRMSE, the poorer the imputation results. This approach is intuitive, but is too restrictive for multimodal data sets: it assumes that there exists a unique answer for a given set of observed variables, which is not true for multimodal distributions. For multimodal data sets, density estimation methods like the Kernel Density Estimation (KDE) (Rosenblatt, 1956; Parzen, 1962) appear of interest for data imputation. But despite some attempts (Titterington & Mill, 1983; Leibrandt & Günnemann, 2018), density estimation methods with missing values remain computationally expensive and not suitable for practical imputation purposes, mostly because they do not generalize well to real-world data sets in spite of an interesting theoretical framework.

Alternatively, other works have developed Gaussian mixture density estimates with Expectation-Maximization (EM) training (Delalleau et al., 2012; McCaw et al., 2020) as well as Gaussian processes for Kernel Principal Component Analysis (KPCA) (Sanguinetti & Lawrence, 2006), but these methods also do not generalize well do heterogeneous numerical data sets in practice. Also, if the mathematical framework of the Missingness Aware Gaussian Mixture Models (MGMM) of McCaw et al. (2020) is interesting, it requires to manually search for the optimal number of Gaussians in the mixture, and is primarily focused on classification tasks. More recently, variants of collaborative filtering algorithms for Matrix Completion problems have been developed (Lee et al., 2016; Li et al., 2020) and can be used for numerical data imputation as well. However, these methods do not seem to perform better than the traditional SoftImpute algorithm (Hastie et al., 2015) for Matrix Completion.

This work focuses on concurrently learning from incomplete data to model and recover missing numerical values. We first look at three simple data sets to illustrate the shortcomings of current data imputation methods with multimodal distributions. We address these issues by introducing a local density estimator that is flexible to accommodate multimodal data structures. By leveraging the convenient properties of the $k$NN-Imputer and the KDE framework, we develop the $k$NN×KDE: a simple yet efficient algorithm for density estimation and data imputation of missing values in numerical data sets.

Using heterogeneous real-world and simulated data sets, we show that our method performs equally or better than state-of-the-art numerical imputation methods, while providing better density estimates for missing values. The code and data used in this work are provided in open-access for the community.

## 2 Problems of current imputation methods with multimodal data sets

In this section, we illustrate problems of current numerical data imputation methods with multimodal data sets. For this purpose, we generate three synthetic data sets in two-dimensional space and qualitatively discuss the imputation performances of four state-of-the-art numerical imputation algorithms with two benchmark methods (column mean and column median).

### 2.1 Three synthetic data sets

The first data set, called `2d_linear`, is a noisy linear distribution. $x_1$ is sampled from a mollified uniform distribution on $[0, 1]$ with standard deviation $\sigma = 0.05$. Then $x_2 = x_1 + \varepsilon$, where $\varepsilon \sim \mathcal{N}(0, 0.1)$.

The second data set, `2d_sine`, is a sine wave with noise. We sample $x_1 = 4\pi u$, where $u$ is drawn from a mollified distribution on $[0, 1]$ with standard deviation $\sigma = 0.05$. Then $x_2 = \sin x_1 + \varepsilon$, where $\varepsilon \sim \mathcal{N}(0, 0.2)$. The noisy surjection allows to show that most imputation algorithms perform well in the unambiguous case (when $x_2$ is missing), but not with multimodal distributions (when $x_1$ is missing).

Finally, `2d_ring` displays a ring with noise. It has been generated in polar coordinates where $\theta \sim \mathcal{U}[0, 2\pi]$ and $r = 1.0 + \varepsilon$, with $\varepsilon \sim \mathcal{N}(0, 0.1)$. Euclidean coordinates are $x_1 = r \cos \theta$ and $x_2 = r \sin \theta$.

These three simple data sets have $N = 500$ observations and are plotted in Figure 1. The code used for generation and the data sets themselves are available on the online repository. We have used a mollified uniform distribution for $x_1$ in `2d_linear` and `2d_sine` to prevent from zero likelihood computation problems at the edges of the uniform distribution.

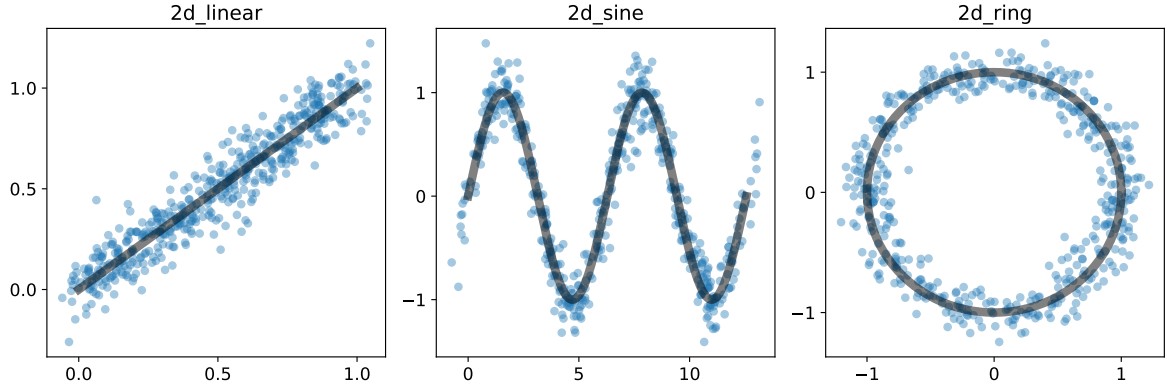

Figure 1: **Three basic synthetic data sets with** $N = 500$ **observations.** `2d_linear` is a bijection, `2d_sine` is a surjection, and `2d_ring` displays a ring and is therefore not a function in the euclidean space.

### 2.2 Five state-of-the-art numerical data imputation methods

Here, we present four data imputation methods used in this work: the $k$NN-Imputer, MissForest, MICE and GAIN. This choice is of course arbitrary, but illustrates well the current state of affairs regarding tabular data imputation (Bertsimas et al., 2018; Poulos & Valle, 2018; Yoon et al., 2018; Jadhav et al., 2019; Woznica & Biecek, 2020; Jäger et al., 2021; Lalande & Doya, 2022; Grinsztajn et al., 2022)

**The $k$NN-Imputer** (Troyanskaya et al., 2001) computes distances between pairs of observations using the NaN-Euclidean distance, which can handle missing values. It imputes missing cells one column at a time by averaging over the $k$ nearest neighbors that have an observed value for the given feature. Therefore, different neighbors can be used to impute various missing entries for the same observation. The hyperparameter $k$ for the number of neighbors is to be optimized.

**MissForest** (Stekhoven & Bühlmann, 2012) is an iterative imputation algorithm. MissForest starts by filling all missing values with initial estimates (typically the column mean), and loops through all columns,

one at a time, performing a regression of that specific column onto all other columns using Random Forests. It stops when the imputed data set is stable enough (following a user-defined threshold) or when a fixed number of iterations has been performed. The number of trees used in the Random Forest algorithm is the hyperparameter to be tuned.

**MICE** stands for Multiple Imputation Chained Equations (van Buuren & Groothuis-Oudshoorn, 2011). Similar to MissForest, it is an iterative imputation algorithm. MICE strictly refers to the algorithmic method which consists of filling missing values using iterative series of regression models one variable at a time. In this work, we use the standard version of MICE that uses linear regressions as a regressor to predict each column successively. This algorithm has no hyperparameter to optimize. MICE has shown good imputation results and is appreciated for its simplicity and absence of hyperparameter tuning, but it fails at capturing non-linear dependencies.

**SoftImpute** is a matrix completion algorithm (Hastie et al., 2015). It works by finding a low-rank approximation of the matrix with missing values while promoting sparsity through a regularization term with coefficient $\lambda$. The algorithm uses an iterative procedure to minimize the objective function. In each iteration, the observed entries of the matrix are used to estimate the missing entries. The estimated entries are then used to update the low-rank approximation of the matrix. This process is repeated until convergence.

Finally, **GAIN** is a GAN artificial neural network tailored for tabular numerical data imputation which claims state-of-the-art numerical data imputation results (Yoon et al., 2018). GAIN smartly revisits the GAN architecture by working with individual cells rather than entire observations. It has recently benefited from a lot of attention for numerical data imputation. However, recent benchmarks show that its performances are mediocre in practice (Jäger et al., 2021; Lalande & Doya, 2022; Grinsztajn et al., 2022). GAIN has several hyperparameters to tune: batch size, hint rate (amount of correct labels provided to the discriminator), number of training iterations, and weight parameter $\alpha$ used in the generator loss.

### 2.3 Imputation results

We introduce missing values in MCAR setting with 20% missing rate. If an observation has both features removed, we repeat the process until at least one feature is present. After missing values have been inserted, we normalize the data set in the range $[0, 1]$ using min/max normalization.

For each data imputation algorithm and for each data set represented as a matrix of size $(N, D)$, we perform a grid search of the hyperparameter than best minimizes the NRMSE:

$$\text{NRMSE} = \sqrt{\frac{1}{N_{\text{miss}}} \sum_{i=1}^{N} \sum_{j=1}^{D} (x_{ij} - \widehat{x}_{ij})^2 \, (1 - m_{ij})} \tag{1}$$

where $m_{ij} = 1$ if cell $(i, j)$ is observed ($m_{ij} = 0$ if missing) and $N_{\text{miss}} = \sum_{i=1}^{N} \sum_{j=1}^{D} (1 - m_{ij})$ is the total number of missing entries in the data set. Imputation results provided by the best hyperparameters are plotted in Figure 2.

Figure 2 provides a concise insight into the current state of numerical data imputation. The scientific consensus is that the $k$NN-Imputer and MissForest overall provide the best numerical data imputation quality, which is somewhat recovered here. MICE uses linear regression between features and cannot capture non-linear dependencies. SoftImpute uses low-rank matrix completion, hence the straight lines as well. Despite its flexible architecture, GAIN performs poorly, even on `2d_linear`. GAIN, like all generative adversarial networks, is difficult to optimize because of training instabilities, mode collapse problems, potential impossibility to converge, or not well defined loss function (Saxena & Cao, 2020).

Both the $k$NN-Imputer and MissForest average over several predictions. This is why the imputation of $x_1$ for the `2d_sine` data set lies between the two sine waves, and imputed values for both $x_1$ and $x_2$ for the `2d_ring` data set are inside the ring. While averaging over several predictions often leads to better estimates, this strategy deteriorates the imputation quality if the missing values distribution is not unimodal.

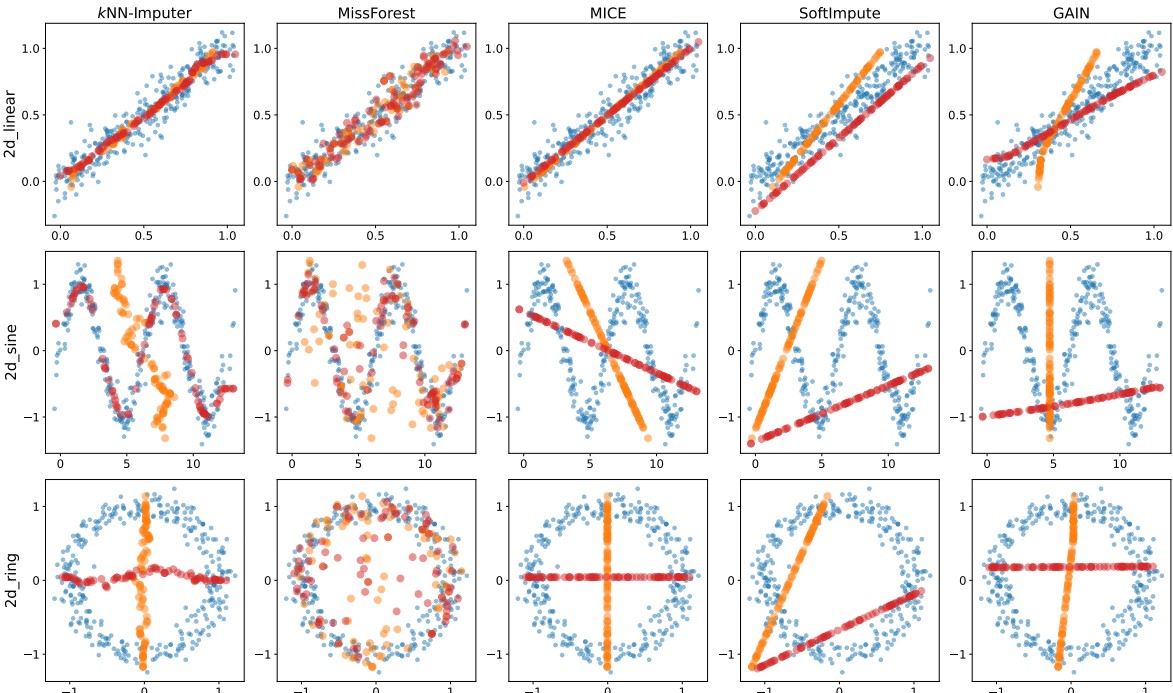

Figure 2: **Imputation results for the three synthetic data sets by the four selected imputation methods with optimized hyperparameters.** Missing data have been injected in MCAR scenario with 20% missing rate. Blue dots correspond to complete observations; orange dots have observed $x_2$ and imputed $x_1$; red dots have observed $x_1$ and imputed $x_2$. The $k$NN-Imputer, MissForest and MICE perform well on `2d_linear`. For `2d_sine`, the $k$NN-Imputer and MissForest can impute $x_2$, but fail at recovering $x_1$. No method can properly impute `2d_ring`.

MICE performs imputation by assuming linear relations between features of the data set. It is therefore no surprise that MICE can very well impute data set `2d_linear`, but fails at imputing data sets `2d_sine` and `2d_ring`. Similarly, SoftImpute uses linear combinations of the observed values as a matrix completion algorithm.

GAIN provides surprisingly disappointing imputation results. While deep-learning models are flexible methods, the generator and the discriminator of GAIN fail to capture the relationship between $x_1$ and $x_2$ in all data sets. Yet innovative, the complex architecture of GAIN (and GANs is general) is problematic to train. This leads to bad imputation results as well as large variability between runs.

## 3 The $k$NN$\times$KDE algorithm

To address the above-mentioned issues related to multimodal distributions, we propose a local stochastic imputation algorithm inspired by the $k$NN-Imputer and kernel density estimation. We adapt the KDE algorithm to missing data settings such that the conditional density of missing features given observed features is estimated.

We use a methodology analogous to the $k$NN-Imputer to look for neighbors, but we work with missing patterns instead of working column by column. The reason of this choice is that working with one column at a time may lead to imputation artifacts as the selected neighbors for various imputed features can be different. Therefore, imputed observations may be incompatible with the original data structure. On the contrary, we are guaranteed to preserve the original data structure if we impute all missing features of an observation at once.

For a data set with $D$ columns, we have up to $2^D - 2$ possible missing patterns. Indeed, each cell may either be missing or not (hence $2^D$ choices) but we do not account for complete cases (nothing to impute) and completely unobserved cases (without even an observed feature).

We first normalize each column of the data set to fit within the range of $[0, 1]$. We refer to this process as the `min-max` normalization. For imputation of the data in row $i$, we compute the distance $d_{ij}$ with all other rows $j$, using the distance

$$d_{ij} = \sqrt{\sum_{k \in \mathcal{D}_{\text{obs}}} (x_{ik} - x_{jk})^2 \ + \sum_{k \in \mathcal{D}_{\text{miss}}} \sigma_k^2} \tag{2}$$

where $\mathcal{D}_{\text{obs}} = \{k \in [\![1, D]\!] \mid m_{ik} = m_{jk} = 1\}$ is the set of indices for commonly observed features in observations $i$ and $j$, $\mathcal{D}_{\text{miss}} = \{k \in [\![1, D]\!] \mid m_{ik} m_{jk} = 0\}$ is the set of indices for features where at least one observation $i$ or $j$ is missing, and $\sigma_k$ is the standard deviation of feature $k$ computed over all observed cells. We call this new distance metric the `NaN-std-Euclidean Distance`, in contrast to the original `NaN-Euclidean Distance` used by the $k$NN-Imputer (Dixon, 1979). See Appendix D for a discussion on this metric properties.

The pairwise distances are then passed to a softmax function to define probabilities:

$$p_{ij} = \frac{e^{-d_{ij}/\tau}}{\sum_j e^{-d_{ij}/\tau}} \tag{3}$$

We use the "soft" version of the $k$NN algorithm, and introduce the temperature hyperparameter $\tau$ which can be interpreted as the effective neighborhood diameter. Instead of selecting a fixed number of neighbors per observation, we consider all observations but give nearest neighbors a stronger weight. In a similar fashion as Frosst et al. (2019), the notion of temperature controls the tightness of each observation's neighborhood. See Appendix A.1 for a discussion on the temperature hyperparameter.

Given a missing pattern, we first select all rows to impute and all the rows corresponding to potential donors. The data to impute is the subset of data which has the current missing pattern, and potential donors are the subset of data where at least all columns in the current missing pattern are observed. For an incomplete observation $i$ in the subset of data to impute, $p_{ij}$ is the probability of choosing observation $j$ from the subset of potential donors. We have $\sum_j p_{ij} = 1$. Algorithm 1 shows the pseudo-code of the $k$NN×KDE.

---

**Algorithm 1:** Pseudo-code for the $k$NN×KDE

**Hyper-parameters:** Softmax temperature $\tau$; Kernel bandwidth $h$; Nb draws $N_{\text{draws}}$

---

**Data:** Incomplete numerical data set $X$
`min-max` normalization in the interval $[0, 1]$;
**for** *each missing pattern* **do**
    $X_{\text{imp}} \leftarrow$ `data_to_impute` $(X, \text{missing pattern})$;
    $X_{\text{don}} \leftarrow$ `potential_donors` $(X, \text{missing pattern})$;
    $d_{ij} \leftarrow$ `NaN_std_Euclidean_Distance` $(X_{\text{imp}}, X_{\text{don}})$;
    $p_{ij} \leftarrow$ `softmax` $(-d_{ij}/\tau)$;
    **for** *each row in $X_{\text{imp}}$* **do**
        $r \leftarrow$ sample $N_{\text{draws}}$ rows from $X_{\text{don}}$ with probabilities $p_{ij}$;
        $e \leftarrow$ sample noise $N_{\text{draws}}$ times from $e \sim \mathcal{N}(0, h)$ with dimension $K$;
        `imputation_samples` $\leftarrow X_{\text{don}}[r] + e$;
    **end**
**end**
`min-max` denormalization;
**Return:** `imputations_samples`

---

The $k$NN$\times$KDE has three hyperparameters: the temperature $\tau$ for the softmax probabilities, the (shared) standard deviation $h$ of the Gaussian kernels, and $N_{\text{draws}}$ the number of imputed samples to draw for each missing cell. The effects of these three hyperparameters are discussed in Appendix A.

For observation $i$ with a missing value in column $k$, the probability distribution of the missing cell $x_{ik}$ is given by

$$p(x_{ik}) = \sum_{j=1}^{N} p_{ij} \, \mathcal{N}\left(x_{ik} | x_{jk}; h\right) \tag{4}$$

where $p_{ij}$ are the softmax probabilities defined in Equation 3, with $p_{ij} = 0$ if observation $j$ is not in the subset of potential donors for observation $i$, and $\mathcal{N}\left(.|\mu; \sigma\right)$ denotes the density function of a univariate Gaussian with mean $\mu$ and standard deviation $\sigma$:

$$\mathcal{N}\left(x | \mu; \sigma\right) = \frac{1}{\sqrt{2\pi\sigma^2}} \, e^{-\frac{1}{2}\left(\frac{x-\mu}{\sigma}\right)^2} \tag{5}$$

If observation $i$ has $K$ missing values, in columns $k_1, k_2, ..., k_K$, then the subset of potential donors will likely be smaller and the joined probability distribution for all missing values is given by

$$p(x_{ik_1}, x_{ik_2}, ..., x_{ik_K}) = \sum_{j=1}^{N} p_{ij} \prod_{\kappa=1}^{K} \mathcal{N}\left(x_{ik_\kappa} | x_{jk_\kappa}; h\right) \tag{6}$$

where the index $\kappa$ runs from 1 to $K$ to denote the successive missing columns, and $p_{ij} = 0$ if observation $j$ is not in the subset of potential donors for observation $i$ like above. As can been seen from Equation 6, the weights $p_{ij}$ are shared such that imputed cells for the same observation have a joined probability that reflects the structure of the original data set.

Note that the pseudo-code of the $k$NN$\times$KDE presented in Algorithm 1 uses $N_{\text{draws}}$ samples for each missing cell. We could instead use the softmax probabilities $p_{ij}$ as weights for the mixture of Gaussians with all potential donors, which would ideally lead to direct probability distributions. We have tried this approach but found that this requires a much larger computational cost, and is only tractable in practice with small data sets. We therefore continue to sample $N_{\text{draws}}$ times to show the returned probability distributions of the $k$NN$\times$KDE.

## 4 Results on the synthetic toy data sets

We show that the pseudo-code of the algorithm presented the proposed method provides imputation samples that preserve the structure of the original data sets. For now, missing data are inserted in MCAR scenario with 20% missing rate, and the hyperparameters of the $k$NN$\times$KDE are fixed to their default values: $h = 0.03$, $1/\tau = 50.0$ and $N_{\text{draws}} = 10,000$.

The upper panels of Figure 3 show the imputation with a sub-sampling size $N_{\text{ss}} = 10$. The sub-sampling size is only used for plotting purposes. If $x_1$ is missing, we sample $N_{\text{ss}}$ possible values given $x_2$ (see the orange horizontal trails of dots), and if $x_2$ is missing, we draw $N_{\text{ss}}$ possible values given $x_1$ (see the red vertical trails of dots).

Of course, it is worth mentioning that if we decide to average over the returned samples by the $k$NN$\times$KDE, then similar artifacts as the ones presented in Figure 2 will arise again. For instance, single point estimates for the `2d_ring` data set will fall inside the ring.

Another way to visualize the imputation distribution for each missing value is to look at the univariate density provided by the $k$NN$\times$KDE algorithm. For each data set, we have selected two observations: one with missing $x_1$ and one with missing $x_2$. The lower panels of Figure 3 show the univariate densities returned by the $k$NN$\times$KDE algorithm with default hyperparameters. The upper left corner of each panel shows the observed value and a thick dashed line indicates the (unknown) ground truth to be imputed. We see that the ground truth always falls in one of the modes of the estimated imputation density.

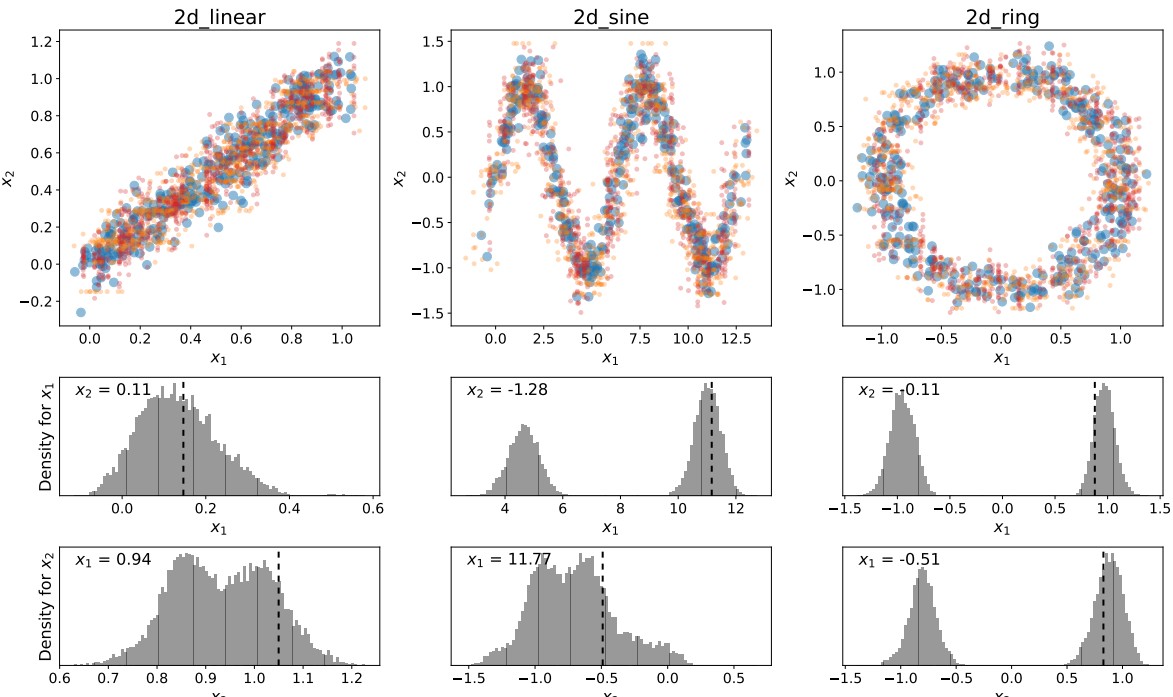

Figure 3: **Imputation results from the $k$NN$\times$KDE algorithm on the three synthetic data sets.** Missing data are inserted in MCAR setting with 20% missing rate. Each missing entry has been imputed by the $k$NN$\times$KDE with default hyperparameters $N_{\mathrm{ss}} = 10$ times for plotting purposes. The imputed values follow the structure of the original data sets. The histograms in the lower panels have $N_{\mathrm{draws}} = 10000$ samples. Thick dashed lines correspond to the (unobserved) ground truth and the observed value is shown in the the upper-left corner. The $k$NN$\times$KDE returns a probability distribution for each missing cell which captures the original data multi-modality structure.

For the `2d_sine` data set, when $x_1$ is missing (central middle panel of Figure 3), the $k$NN$\times$KDE returns a multimodal distribution. Indeed, given the observed $x_2 = -0.88$, three separate ranges of values could correspond to the missing $x_1$. Similarly, the `2d_ring` data set shows bimodal distributions both for $x_1$ or $x_2$, corresponding to the two possible ranges of values allowed by the ring structure.

## 5 Performances on heterogeneous data sets

Now, we assess the practical performances of our method on larger data sets, using both synthetic and real-world data sets from UCI and other repositories. See Appendix B for a comprehensive description of the data sets.

We present imputation results using two metrics: Subsection 5.1 presents the normalized root mean square errors (NRMSE) commonly used for comparing numerical data imputation methods; Subsection 5.2 shows the mean log-likelihood score of the (unknown) ground truth under each imputation model computed over the normalized data in the range $[0, 1]$ for fair comparison. In both cases we test four missing data settings: 'Full MCAR', 'MCAR', 'MAR', and 'MNAR', and six missing rates: 10%, 20%, 30%, 40%, 50%, and 60%. While 'Full MCAR' includes missing data from multiple columns as defined in Section 1, 'MCAR' assumes only one column missing, as in Jäger et al. (2021). See Appendix C for missing data scenario details. For each data set, each missing data setting, and each missing rate, we repeat the imputation `NB_REPEAT=20` times to compute the mean and the standard deviation of the chosen metric.

## 5.1 Imputation results with NRMSE

This subsection presents the imputation results evaluated by the NRMSE, as defined in Equation (1). For the $k$NN-Imputer, MissForest, MICE, the Mean, and the Median imputation schemes, we use the implementation provided by the Python package `sklearn`[2] Pedregosa et al. (2011). For GAIN, we use the original GitHub repository[3] of the authors of GAIN Yoon et al. (2018). As the original package for SoftImpute is in R, we use a more recent Python[4] implementation provided by Muzellec et al. (2020).

When minimizing the NRMSE for a given data set, a given missing data scenario, and a given missing rate, we perform a hyperparameter search except for MICE, Mean, and Median imputation methods, which do not have hyperparameters. We consider the following lists for the other 5 methods' hyperparameters:

- For $k$NN$\times$KDE, the inverse temperature $1/\tau \in [10, 25, 50, 100, 250, 500, 1000]$
- For the $k$NN-Imputer, the number of neighbors $k \in [1, 2, 5, 10, 20, 50, 100]$
- For MissForest, the number of regression trees $N_{\text{trees}} \in [1, 2, 3, 5, 10, 15, 20]$
- For SoftImpute, the regularization term $\lambda \in [0.1, 0.2, 0.5, 1.0, 2.0, 5.0, 10.0]$
- For GAIN, the number of training epochs $N_{\text{iter.}} \in [100, 200, 400, 700, 1000, 2000, 4000]$

When computing the NRMSE for the $k$NN$\times$KDE, we impute with the imputation sample mean.

Tables 1, 2, 3, and 4 show the mean imputation NRMSE for each method and each data set with the missing rate 20%. For each data set, the top three methods that achieve lowest imputation NRMSE have been colored in green, yellow, and orange. We provide the numerical results for the 20% missing rate case as this is often the default missing rate for tabular data imputation benchmarks. The results for all missing rates are available in the online repository for this project.

In order to provide a more concise overview of the imputation NRMSE results, we rank the proposed methods from 1 (best) to 8 (worst) for each data set. For example, looking at the 4th row of Table 1, we have for the `geyser` data set in Full MCAR setting with 20% missing rate: the $k$NN-Imputer (1), the $k$NN$\times$KDE (2), MICE (3), MissForest (4), SoftImpute (5), GAIN (6), Mean (7) and Median (8). Now, for each missing data setting and each missing rate, we compute the mean and the standard deviation for each method ranks over the 15 data sets. Figure 4 shows the average rank for each method.

These results reinforce the previous reports that Deep Learning methods do not perform better than traditional methods on tabular numerical data sets (Bertsimas et al., 2018; Poulos & Valle, 2018; Jadhav et al., 2019; Woznica & Biecek, 2020; Jäger et al., 2021; Lalande & Doya, 2022; Grinsztajn et al., 2022). The proposed $k$NN$\times$KDE consistently achieves the best rank, tightly followed by MissForest. The $k$NN-Imputer and MICE come next. SoftImpute, GAIN, the column Mean, and the column Median are always in the group of the four last methods.

Rankings for MissForest show large error bars because this method is confident in the provided imputation. In other words, the performances of MissForest can vary a lot depending on the nature of the data set (see the standard deviation in the reported NRMSE results in Tables 1 to 4). Alternatively, the $k$NN$\times$KDE and the $k$NN-Imputer have lower rank error bars, indicating that these methods are more consistent across data sets. This can been seen in the lower NRMSE standard deviation in Tables 1 to 4.

It is worth noting that the $k$NN$\times$KDE seems to suffer from the curse of dimensionality, especially in the 'Full MCAR' scenario at high missing rate. Looking at Table 1, the $k$NN$\times$KDE has higher NRMSE compared to other methods for the `breast` and the `sylvine` data sets. Indeed, in 'Full MCAR' scenario at high missing rates for data sets in high dimension, the subset of potential donors for a specific missing pattern can be very low, or even empty therefore preventing from sampling.

While Figure 4 provides the overall ranks, note that the imputation NRMSE can vary greatly between two consecutive ranks. Using again the `abalone` data set NRMSE provided in the first row of Table 1 to

---

[2]`https://scikit-learn.org/stable/modules/impute.html`
[3]`https://github.com/jsyoon0823/GAIN`
[4]`https://github.com/BorisMuzellec/MissingDataOT`

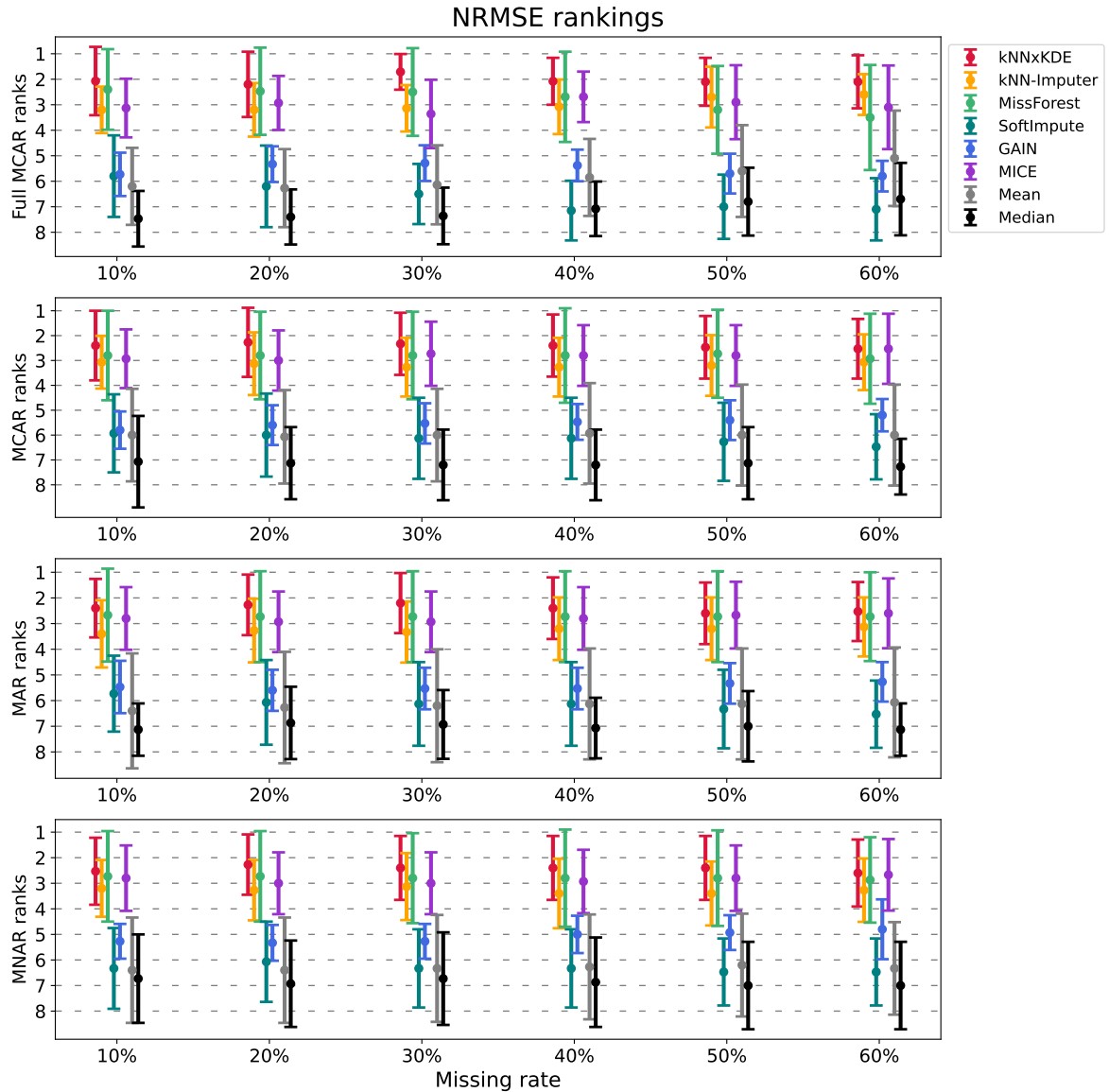

Figure 4: **Average NRMSE rank for each data imputation method in various missing data settings and missing rates.** Our proposed method, the $k$NN×KDE, is consistently the best method, regardless of the missing rate of missing data setting. Second comes MissForest. The $k$NN-Imputer and MICE come next. Besides the column mean or median numerical imputation methods, GAIN and SoftImpute invariably under-perform every other methods.

exemplify, the NRMSE remains below 4.00 for the top four methods, then jumps to 5.29 for GAIN, and finally gets close to 15.00 for the column Mean and Median imputation methods.

Finally, we stress that even though the $k$NN×KDE overall provides minimal NRMSE, the framework used here computes the distribution mean to return a point estimate. Calculating a point estimate brings back the original problem of choosing a single estimate to impute missing data, depicted in Figure 2. For instance, looking at the `2d_ring` data set in Table 1, we see that the $k$NN×KDE does not perform much better than the $k$NN-Imputer or MissForest, which are considered state-of-the-art numerical imputation methods. Therefore, we decide to also measure the performances of the imputation methods with the log-likelihood score.

## 5.2 Performances by log-likelihood score

Next, we look at the log-likelihood of missing values under the probabilistic model provided by each method. For the $k$NN×KDE, a probability distribution for each missing cell is obtained as described in Section 3 and illustrated in Section 4. For the $k$NN-Imputer, we compute the mean and the standard deviation of the $k$ selected neighbors, and calculate the log-likelihood of the ground truth assuming a Gaussian distribution. Similarly for the Mean imputation method, we compute the column mean and standard deviation and assume a Gaussian distribution. For MICE and MissForest, the stochastic nature of these two Iterative Imputer methods allows us to repeat the imputation $N = 5$ times, compute the mean and the standard deviation for each missing value, and calculate the log-likelihood of the ground truth assuming a Gaussian distribution.

Despite being a generative model, GAIN systematically returns a unique value once trained, such that the variability in GAIN's predictions cannot be taken into account. Therefore, we decided not to include GAIN for the likelihood comparative study. We also do not consider the column Median anymore, as we already use the Mean for likelihood computation. We finally discard SoftImpute from this section as well because it showed mediocre performances on the NRMSE rankings and there is no straightforward way to implement a probabilistic version of the SoftImpute algorithm.

When computing the log-likelihood, we do not perform hyperparameter tuning for MissForest, the $k$NN×KDE, and $k$NN-Imputer. Instead, we choose the hyperparameter that best minimized the imputation NRMSE in the previous subsection.

Following the same approach as Subsection 5.1, we present the average log-likelihood scores (computed over all missing cells) for each data set and each method with 20% missing rates in Tables 5, 6, 7, and 8. The numerical results for other missing rates are available online.

In a similar fashion as before, we compute the ranks of the proposed methods using the mean log-likelihood for each missing data scenario and missing rate. For example, looking at the `abalone` data set in Full MCAR mean log-likelihood provided in the first raw of Table 5, we have the following rankings: $k$NN-Imputer (1), $k$NN×KDE (2), MICE (3), Mean (4), and MissForest (5). We average the ranks over all 15 data sets, and present the aggregated results in Figure 5

The $k$NN×KDE provides the overall best mean log-likelihood score, and the $k$NN-Imputer comes next. In 'Full MCAR' missing data setting at high missing rates, the $k$NN-Imputer model returns a higher likelihood score than the $k$NN×KDE. A tentative explanation is that high missing rates in 'Full MCAR' missing setting create sparse observations from which sampling with the softmax probabilities of the $k$NN×KDE can become challenging. In contrast, the $k$NN-Imputer uses independent Gaussian distributions for each column, which may lead to better results when a lot of cells are missing. On a similar note, notice how the column Mean provides greater log-likelihood scores than the MICE algorithm at high missing rates in 'Full MCAR' scenario.

As before, the $k$NN×KDE algorithm can suffer from high missing rates in 'Full MCAR' scenario for high dimensional data sets (see Table 5 for instance) as the subset of potential donors can be small, or even empty. But contrary to the NRMSE case, the log-likelihood score is not as severely affected. Data sets that exhibit a multi-modality structure tend to have much better log-likelihood score results under the $k$NN×KDE probability distribution. These data sets can be identified by looking at the average Dip test $p$-value for the test of unimodality (see Appendix B).

Despite MissForest showing interesting results with the NRMSE as performance metric, it now always scores last when averaging over multiple data sets. This is because the estimates provided by MissForest have a low variability over different runs. As a consequence, the standard deviation used for the Gaussian distribution to model the probability distribution for each missing cell is small, and the resulting shape of the probability distribution is therefore very narrow. In the rare cases where the ground truth falls within $1\sigma$ or $2\sigma$ of the mean provided by MissForest, the likelihood will be high; but in most cases, the ground truth is more than $3\sigma$ away from the MissForest mean, therefore leading to small likelihood of the (unknown) ground truth under the MissForest model.

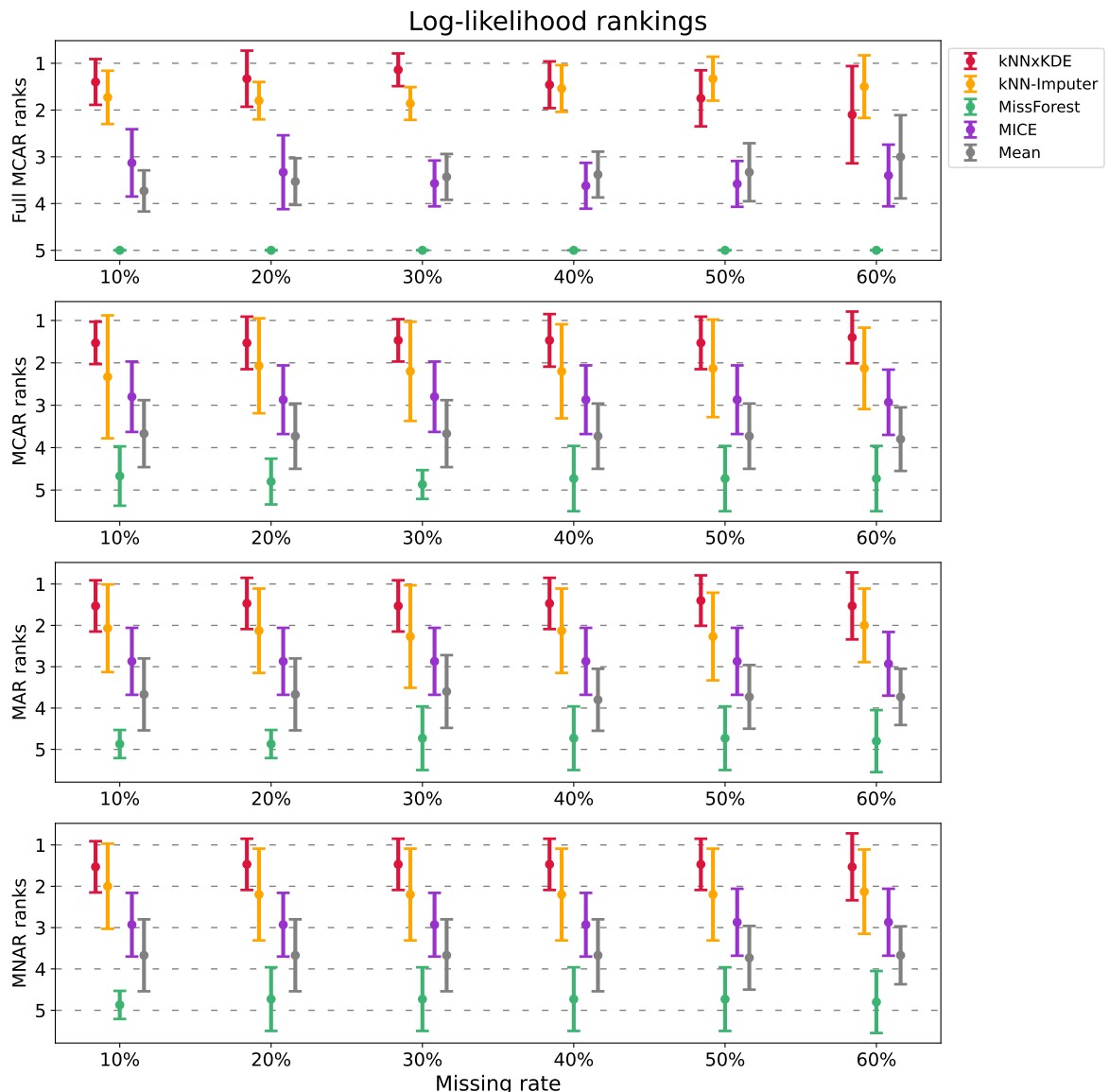

Figure 5: **Average log-likelihood scores rank for each data imputation method in various missing data settings and missing rates.** The $k$NN×KDE ranks best in all cases, except for the Full MCAR scenario at high missing rates, where the $k$NN-Imputer is best. MissForest consistently returns the lowest log-likelihood score, because its predictions do not allow for much variability.

Looking at Tables 5 to 8, we see that for 20% missing rate, the $k$NN×KDE provides the best log-likelihood score, especially for data sets with smaller dimension. As mentioned earlier, both the $k$NN×KDE and the $k$NN-Imputer can suffer from the curse of dimensionality because of the computation of the Euclidean distance in large dimensional spaces.

## 6 Discussion

This work proposes the $k$NN×KDE, a new approach using a soft version of the $k$NN algorithms to derive weights for the Kernel Density Estimation method. The $k$NN×KDE has been developed for numerical data imputation, especially for low dimensional data sets in the presence of multimodality or complex

dependencies. Here, we discuss of the limits and strengths of the $k$NN×KDE, conclude our work, and provide directions for subsequent works.

## 6.1 Limits & Strengths

A substantial drawback is that the $k$NN×KDE becomes computationally expensive in the presence with large data sets. However, it remains faster than MissForest in practice, since it works with missing patterns instead of looping through the data set column by column. This strategy enables to compute only necessary pairwise distances. See Appendix E for quantitative results on computation time.

Another drawback of the $k$NN×KDE is that it cannot impute certain data sets with too many features in 'Full MCAR' and when the missing rate is high. Indeed, in 'Full MCAR' scenario with 60% missing rate for instance, the subset of potential donors (see Algorithm 1) may be empty. In such cases, working on a column-by-column basis, like the standard $k$NN-Imputer, may be an interesting solution.

Now, the great advantage of the $k$NN×KDE is that it preserves the original data structure, which is of major importance when working with multimodal data sets. Our method returns an imputation sample that provides information about the missing data distribution, which is better than a point estimate. By working with missing patterns and imputing all missing features at the same time, the $k$NN×KDE provides a sample of entirely imputed observations that are consistent with the original data set, which is not the case with Iterative Imputation methods (like MissForest and MICE) or the $k$NN-Imputer.

Finally, even though our method consistently achieves the average best imputation NRMSE in all missing data scenarios and at all considered missing rates (see Figure 4), using the sample mean of the returned imputation samples brings the original problem with multimodal distributions back. Looking at the `2d_ring` data set in Table 2, we see that the $k$NN×KDE does not perform better than other methods because of the imputation sample mean. However, we see on Table 6 that the $k$NN×KDE is the only method capable of providing a good density estimation (and therefore a high log-likelihood score) for the `2d_ring` data set. This problem essentially boils down to asking why imputation is needed in the first place: are we interested in subsequent downstream regression or classification tasks ; or are we solely interested in estimating missing values? The common approach of first imputing and then performing downstream tasks may be sub-optimal depending on the choosing imputation strategy (Le Morvan et al., 2021). Instead, the conditional probability distributions returned by the $k$NN×KDE allow to postpone the decision of imputing or not to a later stage. Imputation can subsequently be performed freely: with the mean (to minimize the root mean square error), with the mode (to minimize the absolute mean error), by random sampling (which would prevent from artifacts in the presence of multimodal datasets), or with any other relevant statistic.

## 6.2 Future work

We decided to derive a kernel version on the traditional $k$NN-Imputer, and developed the proposed $k$NN×KDE. Alternatively, it could be interesting to look into another kernel method (or at least any other way to perform density estimation) using Random Forests, since MissForest achieves good results even in its current form.

Another possible extension of this work would be to include an end-to-end treatment of categorical variables within the framework of the $k$NN×KDE. As this study makes use of numerical imputation methods that cannot handle categorical features (e.g. GAIN or SoftImpute), we decided to exclude categorical variables from the scope of this paper. However, tabular data imputation can include numerical and categorical variables in practice and further work may be needed in this direction.

Finally, the `NaN-std-Euclidean` metric appear to yield better results compared to the commonly used `NaN-Euclidean` metric. A possible explanation is that this new metric penalizes sparse observations (with a large number of missing values) by using the feature standard deviation when the entry is missing, therefore preventing to use artificially close neighbours for imputation (see Appendix D). Further investigation of this metric, and experimental results with the standard $k$NN-Imputer may yield interesting insights.

### 6.3 Conclusion

The motivation behind this work was to design an algorithm capable of imputing numerical values in data sets with heterogeneous structures. In particular, multimodality makes imputation ambiguous, as distinct values may be valid imputations. Now, if minimizing the imputation RMSE is an intuitive objective for numerical data imputation, it does not capture the complexity of multimodal data sets. Instead of averaging over several possible imputed values like traditional methods, the $k$NN×KDE offers to look at the probability density of the missing values and choose how to perform the imputation: sampling, mean, median, etc.

Ultimately, this work advocates for a qualitative approach of numerical data imputation, rather than the current quantitative one. The online repository for this work[5] provides all algorithms, all data, and few Jupiter Notebooks to test the proposed method, and we recommend trying it for practical numerical data imputation in various domains.

## Acknowledgments

We would like to thank the three anonymous referees for their time and helpful remarks during the review of our manuscript. In addition, we would like to express our gratitude to Alain Celisse and the good people at the SAMM (Statistique, Analyse et Modélisation Multidisciplinaire) Seminar of the University Paris 1 Panthéon-Sorbonne, for their insightful comments during the development of the $k$NN×KDE.

This research was supported by internal funding from the Okinawa Institute of Science and Technology Graduate University to K. D.

---

[5]`https://github.com/DeltaFloflo/knnxkde`

| | Dim. | kNN×KDE | kNN-Imputer | MissForest | SoftImpute | GAIN | MICE | Mean | Median |
|---|---|---|---|---|---|---|---|---|---|
| 2d_linear | 2 | 7.63 ± 0.39 | 7.72 ± 0.41 | 9.79 ± 0.58 | 11.82 ± 0.77 | 20.73 ± 5.62 | 7.63 ± 0.34 | 24.42 ± 0.91 | 24.49 ± 0.94 |
| 2d_sine | 2 | 18.85 ± 0.94 | 19.21 ± 0.96 | 25.29 ± 1.57 | 43.24 ± 1.97 | 26.82 ± 1.46 | 25.22 ± 0.82 | 26.29 ± 1.11 | 26.36 ± 1.11 |
| 2d_ring | 2 | 29.67 ± 0.83 | 29.77 ± 0.84 | 38.33 ± 1.60 | 42.37 ± 1.39 | 30.21 ± 1.92 | 29.60 ± 0.84 | 29.60 ± 0.84 | 29.74 ± 0.87 |
| geyser | 2 | 10.78 ± 0.93 | 10.77 ± 0.90 | 12.98 ± 1.14 | 17.81 ± 0.97 | 21.95 ± 7.43 | 12.74 ± 0.70 | 29.02 ± 1.10 | 31.34 ± 1.92 |
| penguin | 4 | 12.99 ± 0.65 | 13.64 ± 0.75 | 14.47 ± 0.78 | 24.66 ± 1.54 | 18.37 ± 1.92 | 15.19 ± 0.61 | 22.78 ± 0.83 | 23.28 ± 1.01 |
| pollen | 5 | 10.41 ± 0.21 | 11.82 ± 0.19 | 10.98 ± 0.26 | 17.54 ± 0.28 | 13.61 ± 1.58 | 10.13 ± 0.24 | 14.28 ± 0.21 | 14.28 ± 0.21 |
| planets | 6 | 9.77 ± 0.97 | 11.19 ± 0.77 | 9.16 ± 1.02 | 14.21 ± 1.17 | 12.07 ± 0.89 | 10.22 ± 0.77 | 15.98 ± 0.72 | 17.29 ± 0.91 |
| abalone | 7 | 3.44 ± 0.39 | 3.73 ± 0.36 | 3.32 ± 0.39 | 6.34 ± 0.19 | 5.29 ± 0.70 | 3.86 ± 0.33 | 14.87 ± 0.35 | 14.98 ± 0.35 |
| sulfur | 7 | 5.85 ± 0.16 | 10.01 ± 0.17 | 6.58 ± 0.26 | 14.94 ± 0.14 | 14.13 ± 1.81 | 10.93 ± 0.12 | 17.88 ± 0.14 | 18.90 ± 0.19 |
| gaussians | 8 | 5.47 ± 0.09 | 6.60 ± 0.11 | 5.35 ± 0.11 | 15.38 ± 0.19 | 11.61 ± 1.52 | 9.24 ± 0.12 | 21.94 ± 0.11 | 23.40 ± 0.24 |
| wine_red | 11 | 9.78 ± 0.47 | 10.50 ± 0.44 | 8.58 ± 0.30 | 12.49 ± 0.43 | 12.03 ± 0.61 | 10.08 ± 0.43 | 14.00 ± 0.46 | 14.22 ± 0.47 |
| wine_white | 11 | 8.64 ± 0.42 | 8.75 ± 0.41 | 7.30 ± 0.36 | 11.51 ± 0.37 | 10.81 ± 0.63 | 8.98 ± 0.30 | 11.19 ± 0.54 | 11.28 ± 0.55 |
| japanese_vowels | 12 | 7.97 ± 0.06 | 8.78 ± 0.08 | 7.49 ± 0.11 | 14.32 ± 0.07 | 13.35 ± 0.45 | 11.49 ± 0.08 | 16.18 ± 0.07 | 16.21 ± 0.07 |
| sylvine | 20 | 18.62 ± 0.13 | 18.20 ± 0.14 | 16.98 ± 0.14 | 18.84 ± 0.09 | 19.01 ± 0.29 | 17.64 ± 0.13 | 19.62 ± 0.13 | 20.08 ± 0.14 |
| breast | 30 | 9.17 ± 0.59 | 8.28 ± 0.60 | 6.39 ± 0.54 | 5.79 ± 0.31 | 7.44 ± 0.51 | 5.59 ± 0.32 | 15.29 ± 0.68 | 15.73 ± 0.71 |

Table 1: **Imputation NRMSE (in %) with 20% missing rate in Full MCAR scenario.** MissForest and the kNN×KDE overall perform best at minimizing the NRMSE, followed by MICE and the kNN-Imputer. GAIN does not compete against other numerical data imputation methods.

| | Dim. | kNN×KDE | kNN-Imputer | MissForest | SoftImpute | GAIN | MICE | Mean | Median |
|---|---|---|---|---|---|---|---|---|---|
| 2d_linear | 2 | 6.80 ± 0.49 | 6.83 ± 0.50 | 8.73 ± 0.63 | 12.45 ± 1.39 | 13.81 ± 6.43 | 6.80 ± 0.46 | 21.62 ± 1.56 | 21.66 ± 1.57 |
| 2d_sine | 2 | 6.94 ± 0.57 | 7.26 ± 0.70 | 8.63 ± 0.75 | 42.85 ± 2.77 | 26.13 ± 1.21 | 24.64 ± 1.12 | 26.05 ± 1.13 | 26.09 ± 1.13 |
| 2d_ring | 2 | 29.52 ± 1.42 | 29.64 ± 1.39 | 40.61 ± 2.42 | 43.50 ± 1.10 | 29.54 ± 1.49 | 29.45 ± 1.40 | 29.44 ± 1.40 | 29.47 ± 1.41 |
| geyser | 2 | 10.89 ± 1.21 | 11.01 ± 1.23 | 12.54 ± 1.18 | 21.05 ± 2.04 | 27.72 ± 11.60 | 14.76 ± 1.18 | 33.00 ± 1.27 | 36.18 ± 2.71 |
| penguin | 4 | 9.00 ± 0.93 | 9.20 ± 0.98 | 9.94 ± 0.90 | 13.09 ± 1.31 | 14.09 ± 2.90 | 11.01 ± 1.27 | 23.02 ± 2.04 | 23.55 ± 2.51 |
| pollen | 5 | 4.83 ± 0.24 | 4.92 ± 0.26 | 4.49 ± 0.19 | 15.04 ± 0.42 | 9.11 ± 2.08 | 4.10 ± 0.15 | 14.85 ± 0.64 | 14.85 ± 0.64 |
| planets | 6 | 7.60 ± 0.52 | 7.58 ± 0.57 | 6.97 ± 0.37 | 10.10 ± 0.83 | 9.69 ± 1.18 | 8.23 ± 0.57 | 17.29 ± 0.95 | 18.04 ± 1.35 |
| abalone | 7 | 2.51 ± 0.17 | 2.54 ± 0.17 | 2.54 ± 0.16 | 4.12 ± 0.13 | 4.02 ± 1.00 | 2.60 ± 0.19 | 16.37 ± 0.48 | 16.60 ± 0.53 |
| sulfur | 7 | 1.92 ± 0.09 | 2.02 ± 0.10 | 1.82 ± 0.08 | 8.82 ± 0.16 | 8.29 ± 0.81 | 6.35 ± 0.11 | 20.52 ± 0.24 | 20.57 ± 0.24 |
| gaussians | 8 | 4.69 ± 0.08 | 4.61 ± 0.08 | 4.55 ± 0.09 | 8.02 ± 0.23 | 9.05 ± 1.32 | 6.30 ± 0.11 | 18.98 ± 0.29 | 19.36 ± 0.32 |
| wine_red | 11 | 5.43 ± 0.35 | 6.36 ± 0.32 | 5.04 ± 0.45 | 7.69 ± 0.34 | 8.83 ± 1.07 | 5.48 ± 0.32 | 15.45 ± 0.70 | 15.86 ± 0.71 |
| wine_white | 11 | 5.43 ± 0.65 | 6.21 ± 0.67 | 4.72 ± 0.58 | 8.58 ± 1.46 | 7.97 ± 1.06 | 5.51 ± 1.13 | 8.37 ± 0.89 | 8.38 ± 0.88 |
| japanese_vowels | 12 | 5.34 ± 0.15 | 6.02 ± 0.21 | 6.96 ± 0.11 | 13.26 ± 0.31 | 14.29 ± 0.70 | 10.10 ± 0.14 | 16.79 ± 0.30 | 16.80 ± 0.30 |
| sylvine | 20 | 14.69 ± 0.58 | 14.70 ± 0.57 | 15.21 ± 0.55 | 15.92 ± 0.40 | 15.43 ± 0.69 | 14.62 ± 0.57 | 14.62 ± 0.57 | 14.93 ± 0.64 |
| breast | 30 | 4.39 ± 0.49 | 4.37 ± 0.45 | 0.87 ± 0.32 | 0.75 ± 0.14 | 2.99 ± 0.51 | 0.31 ± 0.04 | 16.69 ± 1.29 | 17.09 ± 1.44 |

Table 2: **Imputation NRMSE (in %) with 20% missing rate in MCAR scenario.**

|  | Dim. | $k$NN×KDE | $k$NN-Imputer | MissForest | SoftImpute | GAIN | MICE | Mean | Median |
|---|---|---|---|---|---|---|---|---|---|
| 2d_linear | 2 | 6.86 ± 0.64 | 6.95 ± 0.66 | 8.66 ± 0.58 | 13.38 ± 0.91 | 18.09 ± 7.46 | 6.82 ± 0.57 | 21.85 ± 1.48 | 22.18 ± 1.50 |
| 2d_sine | 2 | 7.10 ± 0.52 | 7.55 ± 0.54 | 8.85 ± 0.62 | 38.72 ± 2.03 | 26.09 ± 1.63 | 24.40 ± 1.12 | 25.80 ± 1.05 | 25.79 ± 1.06 |
| 2d_ring | 2 | 28.65 ± 1.40 | 28.81 ± 1.45 | 38.57 ± 2.03 | 41.55 ± 2.38 | 28.94 ± 1.64 | 28.56 ± 1.37 | 28.56 ± 1.37 | 28.62 ± 1.39 |
| geyser | 2 | 11.02 ± 1.10 | 11.18 ± 1.11 | 12.38 ± 1.35 | 22.01 ± 1.57 | 26.82 ± 15.20 | 13.91 ± 1.02 | 31.43 ± 1.58 | 30.32 ± 2.90 |
| penguin | 4 | 9.24 ± 0.80 | 9.39 ± 0.75 | 10.18 ± 0.93 | 15.42 ± 2.28 | 15.03 ± 3.73 | 11.30 ± 0.94 | 24.82 ± 2.05 | 26.79 ± 2.45 |
| pollen | 5 | 4.70 ± 0.25 | 4.78 ± 0.23 | 4.41 ± 0.18 | 15.05 ± 0.51 | 8.94 ± 2.11 | 4.07 ± 0.17 | 14.63 ± 0.68 | 14.64 ± 0.68 |
| planets | 6 | 7.40 ± 0.71 | 7.38 ± 0.78 | 6.96 ± 0.63 | 9.95 ± 0.90 | 9.50 ± 1.32 | 7.91 ± 0.73 | 15.90 ± 0.87 | 14.97 ± 0.95 |
| abalone | 7 | 2.59 ± 0.11 | 2.61 ± 0.12 | 2.60 ± 0.12 | 4.08 ± 0.22 | 4.00 ± 1.11 | 2.61 ± 0.13 | 15.97 ± 0.43 | 15.35 ± 0.42 |
| sulfur | 7 | 1.87 ± 0.08 | 1.95 ± 0.09 | 1.78 ± 0.07 | 9.29 ± 0.14 | 8.74 ± 1.53 | 5.79 ± 0.12 | 20.62 ± 0.24 | 21.16 ± 0.24 |
| gaussians | 8 | 4.56 ± 0.08 | 4.49 ± 0.09 | 4.44 ± 0.10 | 8.15 ± 0.19 | 8.61 ± 0.92 | 6.43 ± 0.12 | 17.90 ± 0.29 | 17.60 ± 0.31 |
| wine_red | 11 | 5.02 ± 0.43 | 5.84 ± 0.47 | 4.74 ± 0.48 | 7.37 ± 0.52 | 8.57 ± 1.59 | 5.25 ± 0.34 | 15.07 ± 0.74 | 15.15 ± 0.92 |
| wine_white | 11 | 5.58 ± 0.79 | 6.41 ± 0.84 | 4.74 ± 0.75 | 8.67 ± 1.49 | 8.00 ± 1.09 | 6.07 ± 1.58 | 8.63 ± 1.10 | 8.64 ± 1.11 |
| japanese_vowels | 12 | 5.38 ± 0.16 | 6.08 ± 0.19 | 6.95 ± 0.13 | 12.87 ± 0.28 | 14.46 ± 1.44 | 10.10 ± 0.20 | 16.43 ± 0.27 | 16.47 ± 0.26 |
| sylvine | 20 | 14.70 ± 0.36 | 14.71 ± 0.37 | 15.30 ± 0.40 | 16.06 ± 0.29 | 15.50 ± 0.72 | 14.64 ± 0.36 | 14.64 ± 0.36 | 14.87 ± 0.38 |
| breast | 30 | 4.57 ± 0.40 | 4.62 ± 0.46 | 0.88 ± 0.27 | 0.81 ± 0.22 | 2.92 ± 0.23 | 0.31 ± 0.04 | 17.51 ± 1.28 | 18.36 ± 1.40 |

Table 3: **Imputation NRMSE (in %) with 20% missing rate in MAR scenario.**

|  | Dim. | $k$NN×KDE | $k$NN-Imputer | MissForest | SoftImpute | GAIN | MICE | Mean | Median |
|---|---|---|---|---|---|---|---|---|---|
| 2d_linear | 2 | 6.72 ± 0.65 | 6.74 ± 0.70 | 8.57 ± 0.62 | 13.56 ± 1.29 | 14.92 ± 7.50 | 6.67 ± 0.58 | 21.70 ± 1.70 | 22.08 ± 1.85 |
| 2d_sine | 2 | 7.04 ± 0.68 | 7.28 ± 0.67 | 8.81 ± 0.64 | 48.38 ± 2.19 | 26.43 ± 1.20 | 24.67 ± 1.13 | 26.52 ± 1.04 | 26.96 ± 1.18 |
| 2d_ring | 2 | 29.26 ± 1.16 | 29.41 ± 1.24 | 39.34 ± 2.40 | 50.03 ± 1.67 | 29.42 ± 2.07 | 29.09 ± 1.09 | 29.09 ± 1.09 | 29.42 ± 1.13 |
| geyser | 2 | 10.55 ± 1.03 | 10.65 ± 1.02 | 11.97 ± 1.18 | 22.58 ± 1.57 | 25.92 ± 12.47 | 14.06 ± 1.19 | 32.33 ± 0.98 | 31.33 ± 2.75 |
| penguin | 4 | 9.68 ± 0.95 | 9.84 ± 0.97 | 10.44 ± 0.99 | 16.26 ± 1.74 | 14.93 ± 2.61 | 11.64 ± 1.08 | 24.26 ± 1.70 | 26.33 ± 1.95 |
| pollen | 5 | 4.83 ± 0.28 | 4.90 ± 0.28 | 4.47 ± 0.20 | 17.06 ± 0.73 | 9.38 ± 2.76 | 4.11 ± 0.18 | 15.07 ± 0.68 | 15.08 ± 0.67 |
| planets | 6 | 8.01 ± 0.71 | 8.21 ± 0.84 | 7.57 ± 0.57 | 11.18 ± 0.87 | 9.75 ± 0.73 | 8.59 ± 0.86 | 16.60 ± 0.92 | 15.37 ± 1.07 |
| abalone | 7 | 2.66 ± 0.17 | 2.68 ± 0.18 | 2.67 ± 0.17 | 4.08 ± 0.13 | 3.43 ± 0.35 | 2.68 ± 0.20 | 16.11 ± 0.62 | 15.48 ± 0.59 |
| sulfur | 7 | 1.91 ± 0.14 | 1.98 ± 0.14 | 1.79 ± 0.09 | 9.66 ± 0.13 | 8.28 ± 1.30 | 6.01 ± 0.10 | 20.62 ± 0.18 | 21.27 ± 0.21 |
| gaussians | 8 | 4.41 ± 0.11 | 4.34 ± 0.10 | 4.24 ± 0.10 | 8.67 ± 0.21 | 8.53 ± 1.31 | 5.97 ± 0.14 | 18.90 ± 0.40 | 18.06 ± 0.38 |
| wine_red | 11 | 5.67 ± 0.47 | 6.77 ± 0.42 | 5.26 ± 0.42 | 9.19 ± 0.75 | 9.32 ± 0.94 | 5.92 ± 0.35 | 16.84 ± 0.73 | 18.29 ± 0.72 |
| wine_white | 11 | 6.34 ± 1.01 | 7.21 ± 1.11 | 5.51 ± 0.80 | 9.97 ± 1.60 | 9.47 ± 1.37 | 5.99 ± 1.45 | 9.65 ± 1.36 | 10.04 ± 1.41 |
| japanese_vowels | 12 | 5.59 ± 0.21 | 6.26 ± 0.20 | 7.08 ± 0.19 | 14.76 ± 0.33 | 14.84 ± 2.36 | 10.17 ± 0.24 | 16.99 ± 0.35 | 16.92 ± 0.34 |
| sylvine | 20 | 13.75 ± 0.25 | 13.77 ± 0.27 | 14.54 ± 0.28 | 16.82 ± 0.22 | 14.73 ± 0.77 | 13.65 ± 0.26 | 13.65 ± 0.26 | 12.85 ± 0.34 |
| breast | 30 | 4.61 ± 0.47 | 4.75 ± 0.50 | 1.17 ± 0.44 | 0.96 ± 0.26 | 2.92 ± 0.32 | 0.34 ± 0.05 | 18.59 ± 1.21 | 19.98 ± 1.33 |

Table 4: **Imputation NRMSE (in %) with 20% missing rate in MNAR scenario.**

| | Dim. | $k$NN×KDE | $k$NN-Imputer | MissForest | MICE | Mean |
|---|---|---|---|---|---|---|
| 2d_linear | 2 | 1.13 ± 0.068 | 1.11 ± 0.077 | -5.48 ± 0.589 | 0.65 ± 0.182 | 0.02 ± 0.030 |
| 2d_sine | 2 | 0.89 ± 0.077 | 0.51 ± 0.051 | -4.54 ± 0.445 | -0.64 ± 0.170 | -0.09 ± 0.028 |
| 2d_ring | 2 | 0.28 ± 0.027 | -0.09 ± 0.031 | -5.91 ± 0.348 | -0.82 ± 0.117 | -0.20 ± 0.028 |
| geyser | 2 | 0.81 ± 0.053 | 0.81 ± 0.050 | -10.67 ± 0.432 | 0.12 ± 0.252 | -0.18 ± 0.039 |
| penguin | 4 | 0.67 ± 0.096 | 0.64 ± 0.054 | -4.66 ± 0.343 | -0.03 ± 0.106 | 0.06 ± 0.048 |
| pollen | 5 | 0.94 ± 0.025 | 0.77 ± 0.020 | -3.71 ± 0.102 | 0.62 ± 0.028 | 0.53 ± 0.019 |
| planets | 6 | 1.33 ± 0.090 | 1.98 ± 0.226 | -0.71 ± 0.294 | 0.77 ± 0.057 | 0.55 ± 0.045 |
| abalone | 7 | 2.02 ± 0.019 | 2.22 ± 0.029 | -1.18 ± 0.086 | 1.83 ± 0.040 | 0.64 ± 0.028 |
| sulfur | 7 | 2.04 ± 0.016 | 1.24 ± 0.017 | -0.67 ± 0.058 | 0.69 ± 0.019 | 0.58 ± 0.012 |
| gaussians | 8 | 1.50 ± 0.013 | 1.37 ± 0.011 | -2.87 ± 0.072 | 0.53 ± 0.018 | 0.14 ± 0.010 |
| wine_red | 11 | 1.20 ± 0.022 | 1.05 ± 0.027 | -2.98 ± 0.128 | 0.60 ± 0.062 | 0.66 ± 0.023 |
| wine_white | 11 | 1.34 ± 0.048 | 1.26 ± 0.050 | -2.77 ± 0.081 | 0.89 ± 0.052 | 0.91 ± 0.049 |
| japanese_vowels | 12 | 1.10 ± 0.014 | 1.09 ± 0.007 | -2.10 ± 0.026 | 0.33 ± 0.012 | 0.41 ± 0.005 |
| sylvine | 20 | 0.50 ± 0.009 | 0.47 ± 0.006 | -3.07 ± 0.054 | 0.18 ± 0.013 | 0.32 ± 0.006 |
| breast | 30 | 1.01 ± 0.048 | 1.17 ± 0.034 | -1.51 ± 0.115 | 1.74 ± 0.041 | 0.55 ± 0.023 |

Table 5: **Mean log-likelihood scores with 20% missing rate in Full MCAR scenario.** The average log-likelihood of the missing observations is higher under the $k$NN×KDE density model. The $k$NN-Imputer comes next. MissForest, the $k$NN-Imputer, MICE, or the Mean imputation method are prone to generate artifacts in the imputed data sets, therefore leading to a lower log-likelihood of the ground-truth under their density models.

| | Dim. | $k$NN×KDE | $k$NN-Imputer | MissForest | MICE | Mean |
|---|---|---|---|---|---|---|
| 2d_linear | 2 | 1.19 ± 0.102 | 1.18 ± 0.113 | -8.37 ± 0.359 | 0.81 ± 0.228 | 0.10 ± 0.087 |
| 2d_sine | 2 | 1.11 ± 0.103 | 1.13 ± 0.113 | -8.43 ± 0.454 | -0.62 ± 0.181 | -0.07 ± 0.046 |
| 2d_ring | 2 | 0.32 ± 0.052 | -0.03 ± 0.046 | -8.85 ± 0.490 | -0.69 ± 0.184 | -0.17 ± 0.041 |
| geyser | 2 | 0.82 ± 0.116 | 0.81 ± 0.097 | -11.34 ± 0.176 | -0.11 ± 0.288 | -0.30 ± 0.050 |
| penguin | 4 | 0.85 ± 0.102 | 0.91 ± 0.107 | -4.93 ± 0.572 | 0.34 ± 0.286 | 0.07 ± 0.071 |
| pollen | 5 | 1.53 ± 0.044 | 1.59 ± 0.047 | -3.56 ± 0.229 | 1.28 ± 0.083 | 0.51 ± 0.038 |
| planets | 6 | 1.04 ± 0.143 | 1.09 ± 0.180 | -3.46 ± 0.572 | 0.67 ± 0.161 | 0.35 ± 0.076 |
| abalone | 7 | 2.10 ± 0.025 | 2.30 ± 0.052 | -2.28 ± 0.198 | 1.84 ± 0.086 | 0.38 ± 0.037 |
| sulfur | 7 | 2.36 ± 0.013 | 0.45 ± 0.111 | 0.47 ± 0.129 | 0.89 ± 0.053 | 0.16 ± 0.010 |
| gaussians | 8 | 1.61 ± 0.019 | 1.71 ± 0.023 | -2.97 ± 0.122 | 0.89 ± 0.050 | 0.24 ± 0.020 |
| wine_red | 11 | 1.65 ± 0.075 | 1.26 ± 0.131 | -4.32 ± 0.381 | 0.97 ± 0.109 | 0.44 ± 0.037 |
| wine_white | 11 | 1.77 ± 0.094 | -2.37 ± 0.218 | -4.57 ± 0.426 | 1.18 ± 0.122 | 1.06 ± 0.123 |
| japanese_vowels | 12 | 1.70 ± 0.027 | -0.14 ± 0.106 | -2.04 ± 0.103 | 0.37 ± 0.032 | 0.36 ± 0.018 |
| sylvine | 20 | 0.60 ± 0.026 | 0.50 ± 0.027 | -4.49 ± 0.208 | 0.03 ± 0.069 | 0.51 ± 0.028 |
| breast | 30 | 1.56 ± 0.056 | 1.60 ± 0.133 | 1.26 ± 0.401 | 3.37 ± 0.134 | 0.37 ± 0.041 |

Table 6: **Mean log-likelihood scores with 20% missing rate in MCAR scenario.**

| | Dim. | $k$NN×KDE | $k$NN-Imputer | MissForest | MICE | Mean |
|---|---|---|---|---|---|---|
| 2d_linear | 2 | 1.23 ± 0.068 | 1.19 ± 0.114 | -8.37 ± 0.410 | 0.69 ± 0.213 | 0.10 ± 0.058 |
| 2d_sine | 2 | 1.12 ± 0.068 | 1.11 ± 0.104 | -8.27 ± 0.529 | -0.65 ± 0.204 | -0.09 ± 0.042 |
| 2d_ring | 2 | 0.31 ± 0.043 | -0.05 ± 0.042 | -8.64 ± 0.487 | -0.79 ± 0.197 | -0.18 ± 0.041 |
| geyser | 2 | 0.83 ± 0.098 | 0.82 ± 0.116 | -11.39 ± 0.122 | 0.04 ± 0.196 | -0.26 ± 0.031 |
| penguin | 4 | 0.81 ± 0.125 | 0.92 ± 0.104 | -5.02 ± 0.783 | 0.21 ± 0.259 | -0.01 ± 0.085 |
| pollen | 5 | 1.52 ± 0.046 | 1.57 ± 0.050 | -3.48 ± 0.257 | 1.28 ± 0.080 | 0.48 ± 0.047 |
| planets | 6 | 1.13 ± 0.128 | 1.19 ± 0.118 | -3.59 ± 0.316 | 0.63 ± 0.196 | 0.42 ± 0.043 |
| abalone | 7 | 2.08 ± 0.024 | 2.24 ± 0.035 | -2.40 ± 0.206 | 1.79 ± 0.080 | 0.42 ± 0.027 |
| sulfur | 7 | 2.35 ± 0.012 | 0.43 ± 0.144 | 0.32 ± 0.138 | 1.05 ± 0.052 | 0.16 ± 0.010 |
| gaussians | 8 | 1.63 ± 0.012 | 1.74 ± 0.017 | -2.86 ± 0.144 | 0.82 ± 0.058 | 0.30 ± 0.012 |
| wine_red | 11 | 1.69 ± 0.047 | -2.03 ± 0.357 | -4.32 ± 0.366 | 1.05 ± 0.097 | 0.48 ± 0.042 |
| wine_white | 11 | 1.70 ± 0.147 | 1.04 ± 0.160 | -4.44 ± 0.497 | 1.13 ± 0.186 | 1.00 ± 0.167 |
| japanese_vowels | 12 | 1.67 ± 0.032 | -0.18 ± 0.107 | -2.16 ± 0.098 | 0.36 ± 0.052 | 0.37 ± 0.025 |
| sylvine | 20 | 0.60 ± 0.024 | 0.50 ± 0.028 | -4.46 ± 0.135 | 0.03 ± 0.056 | 0.50 ± 0.028 |
| breast | 30 | 1.53 ± 0.063 | 1.56 ± 0.170 | 1.40 ± 0.433 | 3.42 ± 0.142 | 0.33 ± 0.070 |

Table 7: Mean log-likelihood scores with 20% missing rate in MAR scenario.

| | Dim. | $k$NN×KDE | $k$NN-Imputer | MissForest | MICE | Mean |
|---|---|---|---|---|---|---|
| 2d_linear | 2 | 1.20 ± 0.087 | 1.18 ± 0.118 | -8.24 ± 0.594 | 0.72 ± 0.273 | 0.09 ± 0.080 |
| 2d_sine | 2 | 1.06 ± 0.126 | 1.09 ± 0.135 | -8.25 ± 0.720 | -0.58 ± 0.164 | -0.09 ± 0.056 |
| 2d_ring | 2 | 0.31 ± 0.056 | -0.05 ± 0.054 | -8.54 ± 0.425 | -0.82 ± 0.218 | -0.19 ± 0.048 |
| geyser | 2 | 0.82 ± 0.070 | 0.80 ± 0.093 | -11.39 ± 0.221 | -0.07 ± 0.240 | -0.28 ± 0.040 |
| penguin | 4 | 0.84 ± 0.101 | 0.90 ± 0.113 | -4.43 ± 0.838 | 0.26 ± 0.271 | -0.03 ± 0.113 |
| pollen | 5 | 1.52 ± 0.048 | 1.58 ± 0.051 | -3.51 ± 0.155 | 1.29 ± 0.088 | 0.47 ± 0.039 |
| planets | 6 | 1.05 ± 0.117 | 1.04 ± 0.186 | -3.81 ± 0.595 | 0.63 ± 0.164 | 0.37 ± 0.061 |
| abalone | 7 | 2.09 ± 0.021 | 2.24 ± 0.049 | -2.36 ± 0.234 | 1.80 ± 0.088 | 0.42 ± 0.034 |
| sulfur | 7 | 2.35 ± 0.012 | 0.48 ± 0.136 | 0.39 ± 0.125 | 0.98 ± 0.053 | 0.16 ± 0.007 |
| gaussians | 8 | 1.65 ± 0.019 | 1.77 ± 0.024 | -2.78 ± 0.157 | 0.94 ± 0.040 | 0.24 ± 0.017 |
| wine_red | 11 | 1.61 ± 0.050 | 1.19 ± 0.099 | -4.09 ± 0.405 | 0.96 ± 0.108 | 0.37 ± 0.050 |
| wine_white | 11 | 1.64 ± 0.139 | -2.54 ± 0.264 | -4.62 ± 0.378 | 1.07 ± 0.205 | 0.94 ± 0.176 |
| japanese_vowels | 12 | 1.66 ± 0.031 | -0.26 ± 0.071 | -2.20 ± 0.151 | 0.30 ± 0.044 | 0.35 ± 0.017 |
| sylvine | 20 | 0.70 ± 0.020 | 0.55 ± 0.021 | -4.87 ± 0.150 | 0.10 ± 0.040 | 0.56 ± 0.019 |
| breast | 30 | 1.47 ± 0.075 | 1.43 ± 0.194 | 1.87 ± 0.356 | 3.40 ± 0.167 | 0.25 ± 0.133 |

Table 8: Mean log-likelihood scores with 20% missing rate in MNAR scenario.

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

# A  Discussion on the hyperparameters of the $k$NN×KDE

This appendix section discusses about the qualitative effects for the three hyperparameters of the $k$NN×KDE. We only focus on the 2d_sine data set for visualization purposes. Similar to Figure 3 in the main text, each figure uses a subsample size $N_{ss} = 10$ for plotting purposes, complete observations are in blue, observations where $x_1$ is missing are in orange, and observations where $x_2$ is missing are in red. As before, the subsample size $N_{ss} = 10$ leads to red vertical or orange horizontal trails of points.

## A.1  Softmax temperature $\tau$

Figure 6 shows the imputation quality as a function of the softmax temperature $\tau$, where the Gaussian kernels bandwidth is fixed to $h = 0.03$ and the number of drawn samples is $N_{\text{draws}} = 10000$. For interpretability reason, we report the inverse temperature $1/\tau$.

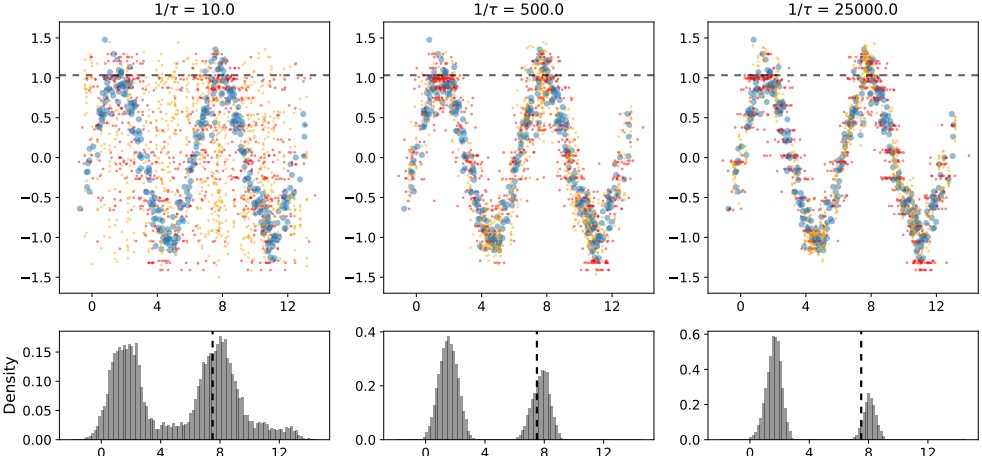

Figure 6: Change in the imputation quality with varying softmax temperature $\tau$. When $1/\tau$ is too low (meaning the temperature is too high), the imputation distribution has a higher variance, hence a large scatter. If $1/\tau$ is too high (meaning the temperature is too low), the imputation distribution will be biased towards the nearest neighbor. The dashed lines show the ground truth coordinates for a randomly selected observation $(x_1, x_2) = (7.50, 1.03)$

On the top-left panel, we see that the neighborhood of each imputed cell is too broad with $1/\tau = 10.0$, resulting in irrelevant neighbors being sampled for imputation and a large scatter. Conversely, the top-right panel shows the imputed samples with an inverse temperature of $1/\tau = 25000.0$ which might be too much and leads to an overfit of the observed data.

The three panels in the bottom focus on a randomly selected observation with with observed $x_2$ and missing $x_1$. In our case, the ground truth is $(x_1, x_2) = (7.50, 1.03)$, and $x_1$ is missing. The vertical dashed line in the upper panels shows the observed $x_2 = 1.03$, and the horizontal dashed line in the lower panels shows the unknown ground truth $x_1 = 7.50$ to be estimated. The lower panels show the histogram for the returned distribution by the $k$NN×KDE for this specific cell. Qualitatively, we see that when the temperature is too high, the imputation distribution is too broad and bridges appear between the two modes of the distribution. On the contrary, when the temperature $\tau$ is too low, the imputation distribution is biased towards the nearest neighbor, resulting in a unimodal distribution. For $1/\tau = 500.0$, the imputation distribution is bimodal which reflects the original data structure, and the ground truth correctly falls in one of the two modes.

## A.2 Gaussian kernels shared bandwidth $h$

Figure 7 shows the imputation quality as a function of the shared Gaussian kernel bandwidth $h$. Here, the softmax inverse temperature is fixed to $1/\tau = 300.0$ and the number of drawn samples is $N_{\mathrm{draws}} = 10000$.

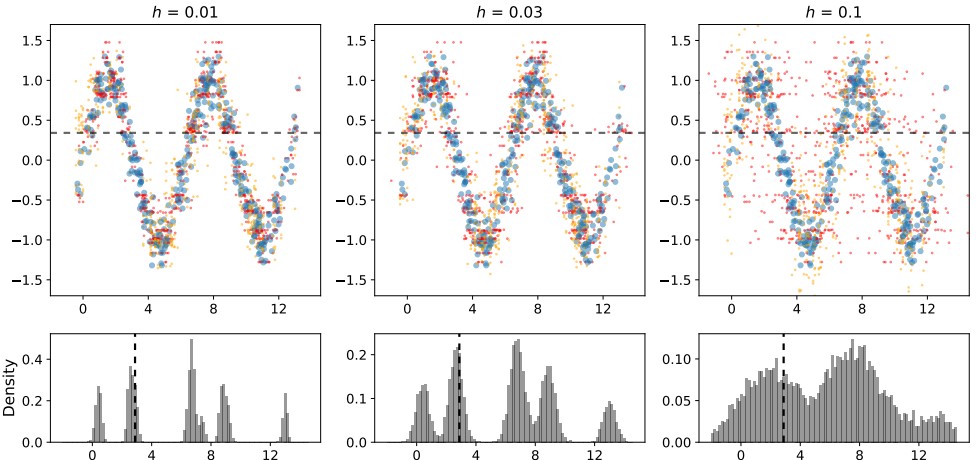

Figure 7: Change in the imputation quality with varying kernel bandwidth $h$. When $h$ is too small, the imputation distribution is sharp. When $h$ is too large, the imputation distribution is blurred. The dashed lines show the ground truth coordinates for a randomly selected observation $(x_1, x_2) = (2.88, 0.34)$

The left two panels show the imputation distribution with a kernel bandwidth $h = 0.01$. A narrow bandwidth results in a tight fit to the observed data, therefore leading to a "spiky" imputation distribution on the bottom-left panel. In the limit where $h \to 0.0$, the returned distribution become multimodal with probabilities provided by the softmax function. On the other hand, the right two panels show the imputation distribution with a large bandwidth of $h = 0.1$, leading to a large scatter around the complete observations.

Like above, the bottom three panels work with a randomly selected observation where $x_2$ is observed and $x_1$ is missing. This time, the ground truth is $(x_1, x_2) = (2.88, 0.34)$. With a narrow bandwidth, we can clearly see the five possible modes corresponding to $x_2 = 0.34$. As the bandwidth becomes larger, modes get closer and eventually merge. The bottom-right panel has only three modes left. In any case, the (unobserved) ground truth $x_1 = 2.88$ always falls in a mode of the imputation distribution returned by the $k$NN×KDE.

Note that the Gaussian kernel bandwidth is shared throughout the algorithm and is therefore the same for all features. As the data set is originally min-max normalized in the $[0, 1]$ interval, the bandwidth adapts to varying feature magnitudes. However, it does not adapt to features scatter, where some features can show a higher standard deviation than others.

Ultimately, we do not optimize the kernel bandwidth in this work. It has been fixed to its default value $h = 0.03$ throughout all experiments in this paper, but could be fine-tuned to obtain higher log-likelihood scores.

## A.3 Number of imputation samples $N_{\mathrm{draws}}$

The imputation distribution returned by the $k$NN×KDE as a function of the number of samples $N_{\mathrm{draws}}$ is shown in Figure 8. Now, the softmax temperature is fixed to $1/\tau = 300.0$ and the Gaussian kernel bandwidth is fixed to $h = 0.03$.

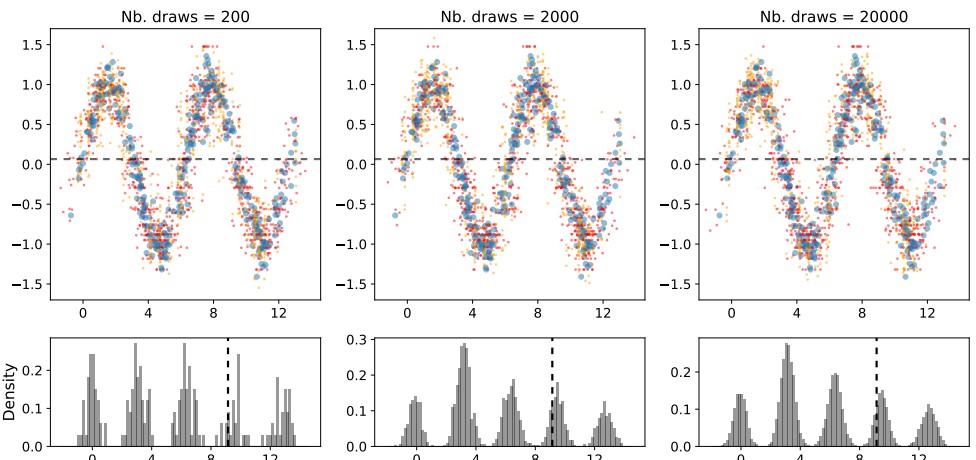

Figure 8: Returned distribution as a function of the number of imputation samples $N_{\text{draws}}$. The bottom three imputation distributions have a higher resolution with larger $N_{\text{draws}}$. The dashed lines show the ground truth coordinates for a randomly selected observation $(x_1, x_2) = (9.15, 0.07)$

In that case, only the number of samples returned by the $k$NN×KDE changes. Therefore, there is no statistical difference between the top three panels besides the random subsamples of size $N_{ss} = 10$ used for plotting purposes.

Like with the hyperparameters $\tau$ and $h$, an observation with observed $x_2$ and missing $x_1$ has been randomly selected. On Figure 8, the ground truth is $(x_1, x_2) = (9.15, 0.07)$. The dashed lines on the top panels show that there are five possible values for $x_1$ given that $x_2 = 0.07$. On the bottom panels, we see that the effect of the number of imputation samples drawn by the $k$NN×KDE defines the resolution of the returned probability distribution. In all cases, the (unobserved) ground truth $x_1 = 9.15$ falls in one of the modes of the imputation distribution.

There is no drawback at setting a large $N_{\text{drawn}}$, besides the obvious computational cost. On the lower-left panel, we see that the value of the probability distribution density can be poorly approximated with $N_{\text{draws}} = 200$. In this work, we always compute the likelihood on the normalized data sets in the $[0, 1]$ interval, and we use 120 evenly spaced bins from $-0.1$ to $1.1$ to allow for outliers. For the imputation with the mean of $k$NN×KDE distributions, we used $N_{\text{draws}} = 1000$ since a high resolution of the imputation distribution is not necessary. However, we used $N_{\text{draws}} = 10000$ when computing the likelihood of the ground truth because a low resolution can lead to likelihood computation error in this case.

## B   Presentation of the data sets

Both real-world and simulated data sets are used in this work. Eleven real-world data sets have been downloaded, most of them from the open access UC Irvine Machine Learning Repository (Dua & Graff, 2019) or the OpenML online repository. Four data sets have been simulated.

This appendix provides summary details about the data. Note that all data sets are complete: they originally do not have missing values. Table 9 presents the data sets name, size, and source. Meaningless rows (like patient ID or row ID) have been removed for data imputation task. Data sets and their description are provided in the online GitHub repository.

In addition, Table 10 provides the mean and standard deviation for the Pearson correlation coefficient, the Spearman correlation coefficient, and the Hartigan Dip Test of unimodality $p$-value (Hartigan & Hartigan, 1985).

Table 9: Name, sizes, and origin of the data sets. Sorted by increasing dimension $D$.

| Data set name | Size | Source |
|---|---|---|
| 2d_linear | (500, 2) | Simulated |
| 2d_sine | (500, 2) | Simulated |
| 2d_ring | (500, 2) | Simulated |
| geyser | (272, 2) | Online (Azzalini & Bowman, 1990) |
| penguin | (342, 4) | Online (Horst et al., 2020) |
| pollen | (3848, 5) | OpenML |
| planets | (550, 6) | NASA Exoplanet Archive (Akeson et al., 2013) |
| abalone | (4177, 7) | OpenML (Nash et al., 1994) |
| sulfur | (10081, 6) | OpenML (Fortuna et al., 2007) |
| gaussians | (10000, 8) | Generated with $k = 4$ random factors |
| wine_red | (1599, 11) | UCI ML (Cortez et al., 2009) |
| wine_white | (4898, 11) | UCI ML (Cortez et al., 2009) |
| japanese_vowels | (9960, 12) | OpenML (Kudo et al., 1999) |
| sylvine | (5124, 20) | OpenML |
| breast | (569, 30) | UCI ML (Bennett & Mangasarian, 1992) |

Given data set of size $(N, D)$ and two columns $k_1$ and $k_2$ in $[\![1, D]\!]$, the Pearson correlation coefficient is defined as:

$$r_{\mathrm{p}}(X_{k_1}, X_{k_2}) = \frac{\sum_{i=1}^{N}(x_{i,k_1} - \bar{x}_{k_1})(x_{i,k_2} - \bar{x}_{k_2})}{\sqrt{\sum_{i=1}^{N}(x_{i,k_1} - \bar{x}_{k_1})^2}\sqrt{\sum_{i=1}^{N}(x_{i,k_2} - \bar{x}_{k_2})^2}}$$

where $\bar{x}_k$ denotes the mean of column $k \in [\![1, D]\!]$.

The Spearman correlation coefficient is obtained by computing the Pearson correlation coefficient over the rank variables. If we denote $R(X_k)$ the rank variable for column $k \in [\![1, D]\!]$, we can write

$$r_{\mathrm{s}}(X_{k_1}, X_{k_2}) = r_{\mathrm{p}}(R(X_{k_1}), R(X_{k_2}))$$

The Pearson correlation coefficient measures the linear correlation between two columns, while the Spearman correlation coefficient quantifies the monotonic relationship (whether linear or not) between two variables. For each data set, we computed both correlation coefficients between all columns, and took their absolute values to compute the mean and standard deviation which we report in Table 10.

As for the Dip Test of unimodality, it tries to assess whether a (univariate) distribution is unimodal or not. The Dip statistic corresponds to the maximum difference between the empirical cumulative distribution function of a sample and the unimodal cumulative distribution function that minimizes that maximum difference. The test computes a $p$-value, which is the probability of obtaining the observed statistic value under the assumption that the distribution is actually unimodal. Lower $p$-values indicate that the distribution of that feature is likely to be multimodal. The last column of Table 10 reports the mean and the standard deviation of the $p$-value computed over the $D$ numerical features for each data set.

### B.1 Simulated Two-Dimensional Data Sets – 2d_linear, 2d_sine, 2d_ring

Three simple data sets in two-dimension are used in this work, named 2d_linear, 2d_sine, and 2d_ring. See Section 2.1 for more details.

### B.2 Abalone Data Set – abalone

This data set is used to predict the age of abalones (a species of marine snails) from physical measurements: sex, shell length, shell diameter, shell height, whole weight, weight of meat, viscera weight, shell weight, and

number of rings which translates to the abalone age (Nash et al., 1994). There are 4,177 observations. The abalone age (target) and the abalone sex have been removed for this work, leading to 7 features.

### B.3   Breast Cancer Wisconsin (Diagnostic) Data Set – `breast`

Ten features are computed from a digitized image of a fine-needle aspiration of a breast mass (Bennett & Mangasarian, 1992). The data set contains 596 observations and 32 columns. The first two columns are removed for numerical data imputation purposes. The other 30 columns are comprised of the mean, the standard error and the mean of the largest three values of the following ten cell features: radius, texture, perimeter, area, smoothness, compactness, concavity, concave points, symmetry and fractal dimension.

### B.4   Simulated Mixture of Gaussians Data Set – `gaussians`

We use a mixture of three multivariate gaussians, generated with random factors method using $k = 4$ factors in dimension $d = 8$. The `gaussians` data set has 10,000 observations and 8 numerical columns.

### B.5   Old Faithful Geyser Data Set – `geyser`

Two features indicate the waiting time between eruptions and the duration of the eruption for the Old Faithful geyser in Yellowstone National Park, Wyoming, USA (Azzalini & Bowman, 1990). This data set is commonly used in machine learning and can easily be found online. It has 272 observations and only 2 features.

### B.6   Japanese Vowels Data Set – `japanese_vowels`

The data has been collected to assess the performances of a multidimensional time series classifier. Nine male speakers uttered two Japanese vowels /ae/ successively. For each utterance, a 12-degree linear prediction coefficients (LPC) analysis is applied, leading to 12 numerical features. Each speaker has various time series for each of its LPC features, which amounts to 9,960 observations.

### B.7   Palmer Archipelago Antartica Penguin Data Set – `penguin`

A total of 342 penguins with 4 features (beak length, beak depth, flipper length and body mass) are organized in 3 classes (Horst et al., 2020). This data set is similar to the famous `iris` data set.

### B.8   NASA Confirmed Exoplanets Archive – `planets`

All confirmed exoplanets according to the NASA Exoplanet Archive as of January 2020 have been downloaded (Akeson et al., 2013). This data set has been generated for a study on exoplanets with the intent to retrieve planetary masses (Tasker et al., 2020). Six planet features have been selected: planet radius, planet mass, planet orbital period, planet equilibrium temperature, host star mass, and number of planets in the system. Only complete observations have been kept, resulting in 550 rows and 6 columns.

### B.9   Pollen Data Set – `pollen`

This is a synthetic data set provided by RCA Laboratories at Princeton, New Jersey. This data set contains 5 geometric features from 3,848 generated pollen grains, namely the length along x-dimension, length along y-dimension, length along z-dimension, pollen grain weight, and pollen density. We could not identify clearly the origin of this data set.

### B.10   Sulfur Recovery Unit Data Set – `sulfur`

The Sulfur Recovery Unit (SRU) is used to remove environmental pollutants from acid gas stream before they are released into the atmosphere. The data set provides 5 variables for 10,081 measures from industrial

processes. These variables describe gas and air flows (Fortuna et al., 2007). The target variables (amount of sulfur) have been removed for imputation purposes.

### B.11 Sulfur Recovery Unit Data Set – `sylvine`

This data set has been generated for a supervised learning data challenge in machine learning where the goal was to perform classification and regression tasks without human intervention. These data has been generated by computing 6 features over a broad variety of other data sets from various domains, which amounts to 5,124 observations.

### B.12 Wine Quality Data Set – `wine_red` and `wine_white`

Typical features (e.g. fixed acidity, citric acid, chlorides, pH, alcohol, ...) have been computed for red and white variants of the Portuguese Vinho Verde wines (Cortez et al., 2009). The red wines data set contains 1,599 observations while the white wines one has 4,989 observations. Both data sets have 11 numerical features, with an additional column indicating the overall wine quality score. This last column has been removed for our imputation work.

Table 10: Mean and standard deviation of Pearson and Spearman correlation coefficients for all data sets.

| Data set Name | Dim. | Pearson correlation | Spearman correlation | Dip Test $p$-value |
|---|---|---|---|---|
| 2d_linear | 2 | $0.95 \pm 0.000$ | $0.952 \pm 0.000$ | $0.605 \pm 0.268$ |
| 2d_sine | 2 | $0.323 \pm 0.000$ | $0.325 \pm 0.000$ | $0.437 \pm 0.436$ |
| 2d_ring | 2 | $0.0117 \pm 0.000$ | $0.014 \pm 0.000$ | $0.000 \pm 0.000$ |
| geyser | 2 | $0.901 \pm 0.000$ | $0.778 \pm 0.000$ | $0.001 \pm 0.001$ |
| penguin | 4 | $0.569 \pm 0.192$ | $0.546 \pm 0.193$ | $0.226 \pm 0.259$ |
| pollen | 5 | $0.297 \pm 0.240$ | $0.287 \pm 0.236$ | $0.953 \pm 0.066$ |
| planets | 6 | $0.501 \pm 0.200$ | $0.534 \pm 0.186$ | $0.390 \pm 0.438$ |
| abalone | 7 | $0.891 \pm 0.058$ | $0.941 \pm 0.031$ | $0.337 \pm 0.398$ |
| sulfur | 7 | $0.239 \pm 0.255$ | $0.235 \pm 0.236$ | $0.303 \pm 0.435$ |
| gaussians | 8 | $0.588 \pm 0.246$ | $0.499 \pm 0.203$ | $0.125 \pm 0.331$ |
| wine_red | 11 | $0.2 \pm 0.187$ | $0.214 \pm 0.193$ | $0.022 \pm 0.047$ |
| wine_white | 11 | $0.178 \pm 0.188$ | $0.194 \pm 0.203$ | $0.010 \pm 0.024$ |
| japanese_vowels | 12 | $0.226 \pm 0.152$ | $0.228 \pm 0.152$ | $0.995 \pm 0.003$ |
| sylvine | 20 | $0.0512 \pm 0.124$ | $0.048 \pm 0.121$ | $0.245 \pm 0.382$ |
| breast | 30 | $0.395 \pm 0.264$ | $0.422 \pm 0.257$ | $0.925 \pm 0.093$ |

## C   Missing data scenarios

We have noticed that the terminology "Missing Completely At Random" (MCAR) is equivocal. In this work, we consider 4 types of missing data scenarios, namely 'Full MCAR', 'MCAR', 'MAR', and 'MNAR'.

In 'Full MCAR', the missing data are inserted completely at random in the entire data set and in all columns. This follows the definition of "MCAR" in the work presenting GAIN (Yoon et al., 2018) for instance.

For 'MCAR', 'MAR', and 'MNAR', we follow the methodology used by the numerical data imputation benchmark of Jäger et al. in which only a single column is masked (Jäger et al., 2021). For this purpose, we select two columns for each data set: the column `miss_col` will be imputed, and the column `cond_col` will be used to compute missing probabilities for the Missing At Random scenario. The selected columns can be seen in the online code. In 'MCAR', missing data are inserted completely at random in column `miss_col`. In 'MAR', we use the quantiles of column `cond_col` to provide conditional probabilities for observations in `miss_col` to be missing. In 'MNAR', the quantiles of column `miss_col` themselves are used to compute missing probabilities.

## D    Study of the new metric

The $k$NN-Imputer makes use of the `NaN-Euclidean Distance` to look for neighbors in the presence of missing data. Given a data set represented as a matrix of shape $(N, D)$, the `NaN-Euclidean Distance` is defined as:

$$d_{ij} = \sqrt{\frac{D}{|\mathcal{D}_{\text{obs}}|} \sum_{k \in \mathcal{D}_{\text{obs}}} (x_{ik} - x_{jk})^2}$$

where $i, j \in [\![1, N]\!]$ are row indices, $\mathcal{D}_{\text{obs}} = \{k \in [\![1, D]\!] \mid m_{ik} = m_{jk} = 1\}$ is the set of column indices for commonly observed features in observations $i$ and $j$ and $|\mathcal{D}_{obs}|$ denotes its cardinality (Dixon, 1979).

The `NaN-Euclidean Distance` is, in essence, a scaled version of the traditional Euclidean distance which compute pairwise distance only when possible. However, this metric can generate artificially small distances when the $|\mathcal{D}_{obs}|$ is low compared to $D$. For example, let us consider $N = 3$ partially observed rows in dimension $D = 5$.

$$\begin{bmatrix} -1 & 6 & 4 & \text{NaN} & 8 \\ \text{NaN} & \text{NaN} & 3 & \text{NaN} & 4 \\ -1 & 5 & 3 & -2 & \text{NaN} \end{bmatrix}$$

Suppose we are interested in estimating the missing value $x_{3,5} = \text{NaN}$. With the `NaN-Euclidean Distance`, the distances are

$$d_{1,3} = \sqrt{\frac{5}{3}(0^2 + 1^2 + 1^2)} \approx 1.83$$

$$d_{2,3} = \sqrt{\frac{5}{1}(0^2)} = 0.0$$

meaning that observations $x_2$ and $x_3$ are at an artificially small distance of $d_{2,3} = 0.0$, and the missing value will most likely be imputed with $x_{2,5} = 4$ rather than with $x_{1,5} = 8$ even though commonly observed cells in rows $x_1$ and $x_3$ appear quite similar. In short, the `NaN-Euclidean Distance` can lead to erroneously small distances between observations with few similar commonly observed features.

To address this problem, we introduce the `NaN-std-Euclidean Distance` defined as:

$$d_{ij} = \sqrt{\sum_{k \in \mathcal{D}_{\text{obs}}} (x_{ik} - x_{jk})^2 \;+\; \sum_{k \in \mathcal{D}_{\text{miss}}} \sigma_k^2}$$

where $\mathcal{D}_{\text{obs}} = \{k \in [\![1, D]\!] \mid m_{ik} = m_{jk} = 1\}$ is as before, $\mathcal{D}_{\text{miss}} = \{k \in [\![1, D]\!] \mid m_{ik} m_{jk} = 0\}$ is the complement of set $\mathcal{D}_{\text{obs}}$, and $\sigma_k$ is the standard deviation for column $k$.

Now suppose that $\sigma_{1,2,3,4,5} = 1.5$, the distances become

$$d_{1,3} = \sqrt{0^2 + 1^2 + 1^2 + \sigma_4^2 + \sigma_5^2} \approx 2.55$$

$$d_{2,3} = \sqrt{\sigma_1^2 + \sigma_2^2 + 0^2 + \sigma_4^2 + \sigma_5^2} = 3.0$$

such that row $x_1$ is now the closest neighbor of row $x_3$.

Using the column standard deviation when pairwise distances cannot be computed allows to penalize observations with a lot of missing values. The standard deviation may be different for each column, which means that missing data for features with a large variance are more greatly penalized that missing data for features with small variance.

While this section introduces the idea behind our new metric with a simple example, we have noticed that it greatly improves the performances of the $k$NN×KDE in practice. Before the change of metric, the NRMSE imputation results of the $k$NN×KDE were closer to the ones from the $k$NN-Imputer, while they are now slightly better than MissForest.

## E   Experimental computation time

For the computational time evaluation, we only focus on the 'Full MCAR' scenario with 20% missing rate. For each data set, and for each numerical data imputation method, we repeat $N = 3$ times the main loop of our algorithm. Results are reported in Tables 11 and 12, along with the data set sizes. These results have been obtained using a 2.3 GHz Dual-Core Intel Core i5 with 16 GB of RAM.

Table 11: Computational time during experiments, in seconds.

| Data set name | Size | $k$NN×KDE | $k$NN-Imputer | MissForest | SoftImpute |
|---|---|---|---|---|---|
| 2d_linear | (500, 2) | 0.281 ± 0.015 | 0.026 ± 0.001 | 0.334 ± 0.026 | 0.744 ± 0.007 |
| 2d_sine | (500, 2) | 0.267 ± 0.013 | 0.026 ± 0.001 | 1.161 ± 0.003 | 0.733 ± 0.005 |
| 2d_ring | (500, 2) | 0.327 ± 0.043 | 0.027 ± 0.000 | 1.172 ± 0.034 | 0.765 ± 0.015 |
| geyser | (272, 2) | 0.153 ± 0.006 | 0.014 ± 0.000 | 0.238 ± 0.029 | 0.667 ± 0.008 |
| penguin | (342, 4) | 0.417 ± 0.013 | 0.029 ± 0.000 | 2.153 ± 0.245 | 0.924 ± 0.036 |
| pollen | (3848, 5) | 7.843 ± 0.115 | 3.056 ± 0.043 | 12.417 ± 0.117 | 4.062 ± 0.005 |
| planets | (550, 6) | 1.062 ± 0.047 | 0.074 ± 0.003 | 3.323 ± 0.824 | 1.417 ± 0.054 |
| abalone | (4177, 7) | 11.962 ± 0.112 | 5.118 ± 0.109 | 10.471 ± 1.222 | 6.066 ± 0.022 |
| sulfur | (10081, 7) | 38.834 ± 0.052 | 36.327 ± 0.579 | 54.081 ± 1.069 | 13.477 ± 0.096 |
| gaussians | (10000, 8) | 42.094 ± 0.195 | 38.594 ± 0.241 | 54.304 ± 8.646 | 15.551 ± 0.093 |
| wine_red | (1599, 11) | 6.744 ± 0.146 | 0.872 ± 0.023 | 20.097 ± 0.105 | 4.328 ± 0.063 |
| wine_white | (4898, 11) | 22.912 ± 0.051 | 10.047 ± 0.096 | 53.790 ± 0.217 | 11.043 ± 0.066 |
| japanese_vowels | (9960, 12) | 60.341 ± 0.092 | 51.194 ± 0.559 | 136.634 ± 0.514 | 24.030 ± 0.073 |
| sylvine | (5124, 20) | 58.984 ± 0.217 | 16.962 ± 0.105 | 178.992 ± 0.157 | 23.380 ± 0.156 |
| breast | (569, 30) | 6.101 ± 0.064 | 0.286 ± 0.004 | 48.564 ± 0.968 | 5.955 ± 0.170 |

Table 12: Computational time during experiments, in seconds. (continue)

| Data set name | Size | GAIN | MICE | Mean | Median |
|---|---|---|---|---|---|
| 2d_linear | (500, 2) | 13.134 ± 0.125 | 0.005 ± 0.000 | 0.000 ± 0.000 | 0.001 ± 0.000 |
| 2d_sine | (500, 2) | 12.885 ± 0.072 | 0.003 ± 0.000 | 0.001 ± 0.000 | 0.001 ± 0.000 |
| 2d_ring | (500, 2) | 13.025 ± 0.040 | 0.003 ± 0.000 | 0.001 ± 0.000 | 0.001 ± 0.000 |
| geyser | (272, 2) | 12.876 ± 0.123 | 0.005 ± 0.000 | 0.000 ± 0.000 | 0.001 ± 0.000 |
| penguin | (342, 4) | 13.174 ± 0.167 | 0.012 ± 0.002 | 0.001 ± 0.000 | 0.001 ± 0.000 |
| pollen | (3848, 5) | 13.793 ± 0.116 | 0.023 ± 0.000 | 0.001 ± 0.000 | 0.002 ± 0.000 |
| planets | (550, 6) | 13.349 ± 0.107 | 0.020 ± 0.000 | 0.001 ± 0.000 | 0.001 ± 0.000 |
| abalone | (4177, 7) | 15.041 ± 0.058 | 0.041 ± 0.008 | 0.001 ± 0.000 | 0.003 ± 0.001 |
| sulfur | (10081, 7) | 16.010 ± 0.142 | 0.054 ± 0.001 | 0.002 ± 0.000 | 0.007 ± 0.000 |
| gaussians | (10000, 8) | 15.443 ± 0.273 | 0.087 ± 0.021 | 0.003 ± 0.000 | 0.008 ± 0.001 |
| wine_red | (1599, 11) | 15.679 ± 0.164 | 0.046 ± 0.001 | 0.001 ± 0.000 | 0.002 ± 0.000 |
| wine_white | (4898, 11) | 16.008 ± 0.121 | 0.095 ± 0.021 | 0.002 ± 0.000 | 0.004 ± 0.000 |
| japanese_vowels | (9960, 12) | 16.681 ± 0.069 | 0.137 ± 0.015 | 0.003 ± 0.000 | 0.011 ± 0.000 |
| sylvine | (5124, 20) | 17.412 ± 0.124 | 0.275 ± 0.094 | 0.002 ± 0.000 | 0.008 ± 0.000 |
| breast | (569, 30) | 18.793 ± 0.068 | 0.208 ± 0.008 | 0.001 ± 0.000 | 0.002 ± 0.000 |

For large data sets, the $k$NN×KDE, the $k$NN-Imputer, and MissForest become time-consuming. MissForest becomes particularly slow when using more than 20 regression trees for data sets where number of total cells (number of variables times number of observations) is large.

## F   Benchmark results on the MovieLens dataset

We tested our method on the MovieLens dataset (Harper & Konstan, 2015), which is known for having inherent missing values with high missing rate. Note that the Matrix Completion problem associated to the MovieLens dataset includes approximately 6,000 users and 4,000 movies with categorical features, and the missing rate associated with this problem is about 96%. The $k$NN$\times$KDE is not designed to work with categorical features, but we could make it work by treating the gender and the age of the users as numerical features and looking for neighbours within users.

We used the leave-one-out cross-validation scheme to assess the performances of the $k$NN$\times$KDE on the MovieLens dataset. We randomly hide one value (corresponding to one random rating for one random user) and use all other users having a rating observed for the randomly selected movie. This process is repeated $N = 100,000$ times to compute the average RMSE. The $k$NN$\times$KDE provides imputation with an average RMSE $= 0.975$. We refer the interested readers to the online repository for the script.

