# OpenReview forum: "Numerical Data Imputation for Multimodal Data Sets: A Probabilistic Nearest-Neighbor Kernel Density Approach"
_TMLR — Accepted by TMLR_

### Review · Reviewer_VbMm · 2023-05-05

**Summary Of Contributions:**

This paper proposes a numerical imputation method that combines KNN with KDE, which can effectively impute missing data with multi-modal distributions. For different missing data patterns, the proposed algorithm uses soft k nearest neighbors of potential donor data to construct a kernel density estimation of the missing data and then draws samples from the density estimation. Compared with existing numerical imputation methods that can only fit unimodal distributions, the proposed method can effectively impute data with complex dependencies and multi-modal distributions. The paper is well-written, logically structured, and easy to follow.

**Audience:**

Yes

**Broader Impact Concerns:**

No broader impact concerns regarding the submission.

**Claims And Evidence:**

Yes

**Requested Changes:**

I suggest that the author refer to my suggestions in weakness to improve the quality of the paper.

**Strengths And Weaknesses:**

Strengths:

This paper proposes a new numerical imputation method from the probabilistic perspective, which can preserve the distribution characteristics of the data and handle different missing data patterns.
The authors have conducted extensive numerical experiments on synthetic toy data sets and heterogeneous data sets to verify the superiority of KNN$\times$KDE over existing numerical imputation methods in imputing missing data with multi-modal distributions.

Weaknesses:

1. In Appendix A.3, the authors indicate that a larger imputation sample size has no disadvantages other than increased computational cost. However, increasing the imputation sample size not only increases computational cost but also leads to the overfitting of regression or classification algorithms since the imputed samples are concentrated around the missing data. The authors need to further discuss the selection of the imputation sample size and its impact on the performance of learning algorithms.
2. In Algorithm 1, the KNN$\times$KDE algorithm depends on many parameters, including the selection of $\tau$, $h$, and $N_{draws}$. The authors minimize the NRMSE to select these parameters. However, NRMSE brings back the original problem of choosing a single estimate to impute missing data and can not reflect the multi-modal distribution of missing data. Therefore, my question is whether there are better rules to select these parameters, such as using the log-likelihood score mentioned in Section 5.2.
3. Some presentation typos:
(i) Algorithm 1 should add a line to represent the input parameters of the algorithm;
(ii) On page 10, line 30, "kNN$\times$" should be changed to "kNN$\times$KDE".

---

> ### Author Response · Authors · 2023-05-12
>
> Dear reviewer,
>
> Thank you for your time and consideration.
> We appreciate the comments and questions you raised. We try to address them in sequential order below.
>
> [1] Large imputation size disadvantages
>
> Appendix A.3. talks about the hyper-parameter $N_{\rm draws}$, and we write that setting a large $N_{\rm draws}$ has no disadvantage, except the obvious computational cost. We understand the point that you raised in your comment: a large imputation sample size can lead to overfitting for downstream regression or classification tasks when we consider multiple imputed datasets.
> However, this does not apply to our evaluation framework. We do not consider multiple imputed datasets. Here, we are interested in the conditional probability density from which we can sample or compute the likelihood of the missing ground truth. In that case, having an imputation sample size $N_{\rm draws}$ as large as possible is only used to control the resolution of the returned probability densities (see the bottom panels in Figure 8). We hope we understood your comment correctly, and that our reply addresses your doubts. If not, could you please elaborate how the imputation sample size might bias the results?
>
> [2] "Are there better rules to select the best hyper-parameters?"
>
> This is a good point... We spent some time discussing the hyper-parameter tuning framework, and decided to adopt the standard RMSE as a metric to minimize in our work. The reason behind this choice is that we wanted the same streamline for all methods, and it would have been unfair to use the log-likelihood to optimize the number of neighbours for the kNN-Imputer, or the number of regression trees for MissForest, as these methods to not provide distributions for the imputed values, but a single point estimate instead.
>
> As your rightly pointed out, minimizing the RMSE brings back the original problem inherent to multimodal distributions. Note that we only optimize $\tau$ in our work, and keep the other two parameters to their default values: $h=0.03$ and $N_{\rm draws}=1000$. With our work, we try to argue that the optimal value of the hyper-parameter $\tau$ does not matter much, compared to the change of imputation method itself: kNN-Imputer, MissForest, GAIN, etc... We have edited the paragraph at the end of Section 5.1. to make this clearer: "Finally, we stress that even though the kNN×KDE overall provides minimal NRMSE, the framework used here computes the distribution mean to return a point estimate. Calculating a point estimate brings back the original problem of choosing a single estimate to impute missing data"
>
> [3] Some typos
>
> Thank you for pointing out some presentation typos. We have corrected them in the new version of the manuscript.
>
> Sincerely yours,

---

> > ### Comment · Reviewer_VbMm · 2023-05-21
> >
> > Thank you for your elaborate responses. I agree with you that the hyper-parameter $N_{draw}$ does not affect the imputation performance, except for the additional computational cost. However, I maintain my opinion that NRMSE is not an appropriate criterion to measure the performance of the proposed $KNN\times KDE$ algorithm to demonstrate its advantages. In Section 5.1, the authors compute NRMSE of $KNN\times KDE$ by taking the distribution mean to return a point estimate, which brings back the original imputation problem. Therefore, I think the experiments in Section 5.1 is not convincing enough to reveal the advantages of the proposed algorithm. I suggest that the authors provide more solid experimental evidence to show the superiority of $KNN\times KDE$ for the multimodal data sets.

---

> > > ### Author Response · Authors · 2023-05-23
> > >
> > > Dear reviewer,
> > >
> > > Thank you for your reply, and we are happy to hear that the confusion regarding the hyper-parameter $N_{\rm draws}$ is now cleared.
> > >
> > > As for the appropriate metric to evaluate the performances of the $k$NN$\times$KDE, we agree with you that the NRMSE between the ground truth and the imputed value is not the best to show the strength of our method, as it brings back the original problem of point imputation with multi-modal numerical datasets. This is why we have two sections used to show the strengths of our method: Section 5.1. shows performances results in terms of NRMSE, and Section 5.2. shows the performances results using the log-likelihood.
> > >
> > > In our manuscript, we show that **not only** the $k$NN$\times$KDE performs overall best when looking at the NRMSE (see Figure 4 in Section 5.1.), **but also** it returns conditional probability densities that capture the original multi-modal distributions of the numerical features (see Figure 5 in Section 5.2.). The results on the log-likelihood show that the missing ground truth has higher likelihood under the probability densities returned by the $k$NN$\times$KDE. We can also see on Tables 5 to 8 that the average log-likelihood under the $k$NN$\times$KDE model is most often the best (green cells in Tables 5 to 8). All other Tables corresponding to different missing rates can be accessed in the Supplementary Materials, and the plots in Figures 4 and 5 can be reproduced using the Jupyter Notebook `rmse_likelihood_results_plots.ipynb`.
> > >
> > > All in all, the proposed method has the *overall* best NRMSE, and MissForest comes next for the NRMSE, and the proposed method also has the *overall* best log-likelihood results, followed by the $k$NN-Imputer. These results strongly indicate that whether we are interested in a point estimate, or we want to capture the multimodal complexity of a numerical dataset with missing values, the $k$NN$\times$KDE works best.
> > >
> > > We hope that our answer correctly addresses your doubts.
> > > Sincerely yours,

---

> > > > ### Comment · Reviewer_VbMm · 2023-05-28
> > > >
> > > > Thank you for your elaborate responses.
> > > >
> > > > We have reached the agreement that the ability of $k$NN-KDE for returning the conditional probability densities that capture the original multi-modal distributions of the numerical features has been proved by the experiments of the log-likelihood evaluations.
> > > >
> > > > Meanwhile, I agree with the experiments in Section 5 Figure 3 that the missing value locates around one of the localized modes in the conditional distribution. However, I am still confused about the single-point imputation part if the conditional distribution obtained by the $k$NN-KDE algorithm is multi-modal, that is, how can I determine which modes should I choose to be the imputation value?
> > > >
> > > > In fact, the experiments in Section 5 Figure 3 with the NRMSE evaluation still confuses me based on the following facts:
> > > > 1. Using the distribution mean of the estimated conditional distribution obtained by $k$NN-KDE, as stated in Section 4, can produce "similar artifacts presented in Figure 2".
> > > > 2. the NRMSE in Figure 3 is calculated with the imputation value defined as the sample mean.
> > > > Take the 2d_ring as an example, the last 2 rows in Figure 3 demonstrate the conditional distribution of the chosen point, and if we calculate the sample mean, high chances are that the imputation value will locate around $0$. If we plot the scatter plot of the imputation result, I guess the plot can ensemble the 2d_ring result of MICE in Figure 2.
> > > >
> > > > To summarize, the experiments of the log-likelihood measure for evaluating the estimated conditional distribution are satisfactory and convincing. Nevertheless, for the single-point imputations, I suggest the authors provide more solid experimental details to diversify the $k$NN-KDE.

---

> > > > > ### Author Response · Authors · 2023-05-29
> > > > >
> > > > > Dear reviewer,
> > > > >
> > > > > Thank you for taking the time to read our reply and come back to us. We appreciate that you shared your new interrogations with us. Below, we try to elucidate your confusion and refer to the appropriate part of the manuscript, which we have edited.
> > > > >
> > > > > Everything you have stated is correct, and your confusion regarding the NRMSE probably comes from the fact that we have two frameworks to evaluate the performances of the imputation methods. (i) The NRMSE measures the error between imputed value and ground-truth (see Section 5.1, and Figure 4), and (ii) the log-likelihood quantifies how the ground-truth fits under each model (see Section 5.2. and Figure 5).
> > > > >
> > > > > When we compute the NRMSE, we need to compute a point estimate. We chose to impute with the conditional density mean. As you correctly pointed out, this approach brings back the original problem of falling between distribution modes. Indeed, with this approach, the imputed values from the $k$NN$\times$KDE would fall inside the `2d_ring` dataset, like the standard $k$NN-Imputer. In this regard, we refer you to Tables 1 to 4, where the results regarding the NRMSE for the $k$NN$\times$KDE is not necessarily better than the NRMSE results for the $k$NN-Imputer or MissForest on the `2d_sine` and `2d_ring` datasets.
> > > > >
> > > > > However, unlike the $k$NN-Imputer (or any other traditional method covered by this study), the $k$NN$\times$KDE returns a **probability distribution** from which we can sample. This is what is depicted on Figure 3 with the conditional distributions on the bottom panels. For the log-likelihood results, we refer to Tables 5 to 8: we can see that the $k$NN$\times$KDE does better than other imputation methods.
> > > > >
> > > > > In order to diversify the $k$NN$\times$KDE as you suggest, we could instead choose to impute by randomly drawing one imputed sample, or by selecting the mode of the returned conditional probabilities. These approaches would prevent from generating artifacts when choosing a point estimate. However, these approaches lead to worse NRMSE than the standard mean. If we used the **absolute** mean error, then the mode would provide the lowest error.
> > > > >
> > > > > In light of your confusions and questions, we have edited Section 6.1. (limits and strengths), and we copy-paste here the part we believe to be the most relevant:
> > > > > > This problem essentially boils down to asking why imputation is needed in the first place: are we interested in subsequent downstream regression or classification tasks ; or are we solely interested in estimating missing values? The common approach of first imputing and then performing downstream tasks may be sub-optimal depending on the choosing imputation strategy (LeMorvan2021). Instead, the conditional probability distributions returned by the $k$NN$\times$KDE allow to postpone the decision of imputing or not to a later stage. Imputation can subsequently be performed freely: with the mean (to minimize the root mean square error), with the mode (to minimize the absolute mean error), by random sampling (which would prevent from artifacts in the presence of multimodal datasets), or with any other relevant statistic.
> > > > >
> > > > > We hope that these additional comments can elucidate your confusion, and we remain at your disposal may you have any other concern.
> > > > >
> > > > > Sincerely yours,

---

### Review · Reviewer_RUkZ · 2023-05-08

**Summary Of Contributions:**

The authors present an imputation approach for numerical tables based on a combination of a kNN Models and a kernel density estimate of the to-be-imputed attribute, conditioned on the observed attributes.

Extensive empirical comparisons with other established imputation methods investigate the performance of the proposed approach under a variety of missingness patterns and conditions.

The strongest parts of this work in my opinion are the extensive experiments under a variety of missingness conditions and solid comparisons with many other methods. The empirical evaluation is very convincing - it could be strengthened further though if the authors would have considered tables with categorical / ordinal attributes as well.

Another positive aspect is the simplicity of the proposed approach.

There were a couple aspects that could be improved, but the main point on that list is the empirical comparison. While the empirical evaluations were done very well, the results do not seem to be very convincing overall. And the results on the synthetic data could be explained a bit better - in my experience kNN and random forests should yield a better imputation result on those nonlinear tasks in fig. 2 and it is not clear how and why the proposed approach performs better in these two scenarios.


**Audience:**

Yes

**Broader Impact Concerns:**

If imputation methods are easy to use, as this one here, other researchers will use it on any data which can lead to potentially wrong or biased imputations. If these imputations will be taken as real data in downstream analyses, this could lead astray conclusions of researchers who are not aware of the imputation operations performed.

**Claims And Evidence:**

No

**Requested Changes:**

- include data sets with categorical attributes
- Explain synthetic data results better
- Better illustration of the empirical performance of the proposed approach (maybe scatterplots between proposed method and RF / kNN to better assess visually that the proposed approach really is consistently better?)

**Strengths And Weaknesses:**

# Strengths

+ The idea to use density estimates on well performing existing imputation algorithms is attractive

+ The approach proposed is appealing in its simplicity. In fact I believe it would be interesting to simplify it further and apply that principle also to other imputation methods, such as forests or other ensemble methods, replacing the inner sampling loop in algorithm 1. That way, one could also try to use the proposed method as a ‚base estimator‘ for MICE?

+ The authors selected kNN as the base imputation algorithm in their proposed solution, which has been shown to work well for many tabular data sets

+ The experimental evaluation on a variety of data sets is very well designed and implemented

+ Probing the algorithm under various missingness conditions and missingness ratios helps to demonstrate the empirical performance comprehensively

+ the idea of drawing bootstrap samples in the inner loop of the proposed algorithm is convincing - especially considering the scaling problems associated with KNN

# Potential improvements

- minor: the title contains multimodal and this often refers to combinations of different data sources, images, text, tables. Here it refers to multimodal distributions, right?

- The empirical comparisons do not appear to strongly support the conclusion that the proposed method performs better than established imputation techniques such as MissForest or kNN. The results in table 1 through 8 indicate that the imputation performance is overall not substantially improved.

- The synthetic data examples shown in fig. 2 could be explained a bit better; my intuition from very similar experiments is that the kNN and the random forest should be able to capture the conditional distributions of one column conditioned on the other column well. If the imputations in fig. 3 are computed with i.e. more training data, other missingness patterns or generally different experimental conditions as those in fig. 2, this could be highlighted. As motivating examples it would be helpful if fig. 2 demonstrated more clearly how exactly the proposed solution solves the problem that e.g. the kNN imputation has and why the gaussian density assumption would help on those orange dots, that are pretty far off from the ground truth.

- The study considers only numerical data. Most of the relevant related literature on tabular data imputation considers at least also categorical data, some also ordinal data. It would strengthen the study to include at least some data sets with more heterogeneous data types. This complicates the analysis of the empirical results, but as the relevant figures are based on rank statistics, also imputation based on classification methods would be simple to include - and most methods compared with support that sort of data out of the box.

- One reason why kNN imputations have not been adopted in practice as much as other solutions is their poor scalability to large data sets. I might have missed that, but if not the authors could mention that limitation. The bootstrap sampling in the inner loop of algorithm 1 could address this problem but it is not entirely clear how this approach would trade off imputation quality with N_{draws}.

- the bootstrap sampling in algorithm 1., inner loop, could be / is part of other imputation implementations, e.g. forest based ones, too. It would be interesting to see how much of the improvements that were observed in some cases for the proposed method are due to this resampling. And whether existing other methods could improve with that, as well. I think that would strengthen the proposed method, if it’s applicable and tested for other ‚base‘ models than kNN, too.

---

> ### Author Response · Authors · 2023-05-15
>
> Dear reviewer,
>
> Thank you for your time and your insightful comments which we carefully read. Below, we provide potential answers and additional comments following your sequential order.
>
> -- Strengths --
>
> [2, 3] The proposed method uses weighted kernels around the nearest neighbours of a missing observation. As you pointed out, this idea could be extended to other imputation methods like MICE or MissForest. However, generalization would not be as straightforward as replacing the inner sampling loop of Algorithm 1. We chose to implement our approach as a kernel extension of the kNN-Imputer, because we can compute weights using NaN-Euclidean distance and softmax, in a similar fashion as the original kNN-Imputer. We have also thought about potential kernel extensions for MissForest as well, but the framework would be different and new questions arise: How do we weight regression trees? What is a satisfactory number of trees?... Therefore, our work is limited to this new version of the kNN-Imputer (which we called kNNxKDE), and other imputation algorithms could be worth being investigated in the future. We added a new paragraph (Section 6.2) to talk about potential future directions.
>
> -- Potential improvements --
>
> [1] Yes, the world "multimodal" here refers to statistical distributions of numerical random variables having more than one mode. In order to clear from the confusion with "multimodal" input data (images, video, text, sound, ...), we explicitly used "Numerical Data Imputation" in the title of the manuscript.
>
> [2] Tables 1 to 8 are a sub-sample corresponding of to 20% missing rates only (default missing rate in most research studies). These tables are used for illustration. All tables are available in the Supplementary Materials, and we provide the code to re-generate them. In our work, we do not argue that the kNNxKDE **consistently** provides the best imputation as measured by the NRMSE. For certain datasets, MissForest or the standard kNN-Imputer provide better NRMSE results than the kNNxKDE. But we try to show with Figure 4 that the kNNxKDE provides the **overall** best imputation method when looking at a large number of datasets across various missing rates. We agree with you that showing aggregated results is challenging with 7 imputation methods (now 8), 4 missing scenario, 6 missing rates and 15 datasets. We decided to come up with the scoring system (Figures 4 and 5) to display overall results, and all results are provided in Supplementary Materials for further investigation.
>
> Also, we think that the main strength of our method is that it returns a probability distribution that correctly capture the underlying probability distribution of the original dataset, as shown by the ground truth likelihood results (Section 5.2 and Figure 5). Note that MissForest which was the overall second best imputation method as measured by the NRMSE, is now the worst at providing reliable density estimates for the ground truth (due to its bias-variance trade-off in favour of low variance). We edited a paragraph in Section 5.2. to make it clearer.
>
> [3] The results of Figure 3 do **not** use more training data or a different setting than the one used in Figure 2. We encourage you to experiment with the Python notebook `make_other_figures.ipynb` provided in Supplementary Materials, which allows to very quickly reproduce Figure 2 with `sklearn` Python package. As MissForest and the kNN-Imputer aggregates their individual predictions (from individual neighbours or individual trees) to provide a point estimate for imputation, their imputed values fail to capture bi-modal distributions. Our method proposes not to aggregate in order to take into account bimodality. When introducing Figure 3 in the manuscript, the main text says that "it is worth mentioning that if we decide to average over the returned samples by the kNN×KDE, then similar artifacts as the ones presented in Figure 2 will arise again. For instance, single point estimates for the 2d_ring data set will fall inside the ring."
>
> [4] At this stage, categorical and ordinal variables are beyond the scope of our study because some numerical imputation algorithms (like MICE, GAIN, and the newly added SoftImpute after following reviewer #1 comments) cannot work with categorical data. The kNNxKDE can easily be extended to categorical variables. We have been testing it on generated datasets, but we haven't yet tested it with large scale real-world datasets including categorical features. We added a paragraph in Section 6.2 regarding categorical features the kNNxKDE.
>
> [5] We decided to repeat a subset of our experiments and store the computation time during training. We added the new Appendix E that compares the computational time of the various algorithms used in this work, and added some comments in Section 6.1.
>
> [6] See the discussion in "Strength".
>
> We remain at your disposal may you have any other questions.
>
> Sincerely yours,

---

### Review · Reviewer_PGgQ · 2023-05-11

**Summary Of Contributions:**

This paper proposes an imputation method called kNN-KDE and demonstrates its effectiveness in various simulated datasets and real datasets with simulated missingness.

**Audience:**

Yes

**Claims And Evidence:**

No

**Requested Changes:**

Please address the weaknesses that I mentioned.

**Strengths And Weaknesses:**

Strengths:
- overall I found the paper easy to follow
- the proposed method is simple and easy to understand

Weaknesses:
- no comparison to methods commonly used for matrix completion, which is also doing imputation such as SoftImpute (Hastie et al. 2015); there's also a somewhat recently developed nearest-neighbor-inspired algorithm that can be used for imputation by Lee et al 2016/Li et al 2019; I think commenting on how the proposed method kNN-KDE compares with these matrix completion/collaborative filtering approaches would be helpful
- no experiments on real datasets with real missingness (for example, in collaborative filtering literature, it's common to use MovieLens/Netflix/Yahoo datasets, which essentially have MNAR missingness)
- I think providing more theoretical analysis of the proposed method would be helpful to give a better sense of when and why we should expect it to work well
- I think it's important to also consider what imputation is even being used for. Often it is used as just a preprocessing step in which the "real" problem that might be of interest is some downstream prediction task. In this case, there is already a fairly strong theoretical result on what a "good" imputation even is (Morvan et al 2021). I think providing commentary on this would also be helpful.

References:
- Trevor Hastie, Rahul Mazumder, Jason D Lee, Reza Zadeh. "Matrix completion and low-rank SVD via fast alternating least squares". JMLR 2015.
- Christina E Lee, Yihua Li, Devavrat Shah, Dogyoon Song. "Blind Regression: Nonparametric Regression for Latent Variable Models via Collaborative Filtering". NeurIPS 2016.
- Yihua Li, Devavrat Shah, Dogyoon Song, Christina Lee Yu. "Nearest neighbors for matrix estimation interpreted as blind regression for latent variable model". IEEE Transactions on Information Theory 2019.
- Marine Le Morvan, Julie Josse, Erwan Scornet, Gael Varoquaux. "What’s a good imputation to predict with missing values?". NeurIPS 2021.

---

> ### Author Response · Authors · 2023-05-19
>
> Dear reviewer,
>
> Thank you for taking the time to provide valuable comments and recommendations on our manuscript. We have addressed your points and provide comments and answers below in sequential order.
>
> [1] Lack of comparison with Matrix Completion methods
>
> We indeed overlooked the SoftImpute algorithm, and we are grateful that you brought it to our attention. We took the time to add the SoftImpute Matrix Completion algorithm in our benchmark and re-run all our experiments. Because SoftImpute is originally implemented in R, we used the Python implementation provided by Muzellec et al. in their paper Missing data imputation using optimal transport (2020). We provide the new version of our code in Supplementary Materials.
>
> We have updated the ranking provided in Figure 4 of our manuscript. Taking into account all missing data scenario, missing rates, and data sets, there seems to be now two groups: best four methods (kNNxKDE, kNN-Imputer, MissForest, MICE), and worse four methods (GAIN, SoftImpute, Mean, Median). We have added few new paragraphs in Sections 2, 5, and 6 to to highlight and explain the results provided by SoftImpute. We have also updated Figure 2 which now contains an additional column to show the results of SoftImpute on the three simple synthetic data sets.
>
> We have also added some comments in the main text regarding the references you pointed at: "Blind Regression: Nonparametric Regression for Latent Variable Models via Collaborative Filtering", Lee et al. NeurIPS 2016, and "Nearest neighbors for matrix estimation interpreted as blind regression for latent variable model", Li et al. IEEE Transactions on Information Theory 2019.
>
> [2] No experiments on real data sets with real missingness
>
> The final goal is to apply our method to another problem in Astrophysics which inherently has missing values (like the MovieLens dataset or the Netflix dataset). That said, datasets with real missing values have been intentionally removed from our study because we do not have a way to assess the performances of the various data imputation methods without the ground truth values. To the best of our knowledge, all data imputation benchmarks that we have seen (Bertsimas2018, Poulos2018, Yoon2018, Jadhav2019, Woznica2020, Jager2021, Lalande2022, Grinsztajn2022) include complete datasets to assess for the sake of comparison.
>
> [3] More theoretical analysis
>
> This manuscript was first submitted to JMLR, and the Action Editor of JMLR recommended us to submit this manuscript to TMLR instead, due to the lack of a thorough theoretical analysis of the proposed algorithm. While we do not provide theoretical analysis for why and when our algorithm works, we have added experimental insights in Appendix. Appendix A explains how the hyper-parameters affect the output probability distribution of the kNNxKDE: Appendix A.1 shows how the softmax temperature affects the weights and discuss what happens in the limit when $\tau \rightarrow +\infty$ and $\tau \rightarrow 0$ ; and Appendix A.2 provides the same analysis with $h \rightarrow 0$ regarding the bandwidth. Additionally, Appendix D presents the newly introduced metrics (called NaN-std-Euclidean distance) and qualitatively explains why it gives more meaningful distances than the previous NaN-Euclidean distance used by the standard kNN-Imputer. We also added the new Section 6.2. where we present further possible work directions, especially to investigate the properties of the NaN-std-Euclidean distance.
>
> [4] Discuss why imputation is even needed in the first place
>
> Thank you for bringing the results of the paper "What’s a good imputation to predict with missing values?" to our attention. Based on your comments and the insights provided by the paper, we added a new paragraph in the main text (at the end of Section 6.1) to discuss what do to with imputation and possible downstream tasks.
>
> You can find on top of this page the new version of our manuscript. We hope that we have addressed your main concerns, and remain at your disposal to answer further questions.
>
> Sincerely yours,

---

> > ### Comment · Reviewer_PGgQ · 2023-05-19
> > **Thanks for the response!**
> >
> > Thanks for addressing the bulk of my concerns.
> >
> > I am still inclined to suggest adding experiments on data with real missingness. The MovieLens/Netflix datasets are very much standard and there are standard evaluation metrics used for them for many, many years now *despite* them not being complete datasets (roughly, the evaluation is based on withholding some fraction of revealed entries and looking at how well the imputation filled in these particular entries that we do have ground truth for; for the Netflix dataset in particular, there is already a pre-defined set of ratings to treat as the "test" data). I think readers would find it valuable just to see how your proposed method kNNxKDE compares in such settings to existing methods. There is also the Yahoo R3 dataset where the test set is (unlike the training set) intentionally set up to have MCAR missingness.
> >
> > The rest of your response looks reasonable to me.

---

> > > ### Author Response · Authors · 2023-05-23
> > > **Experiment results with the MovieLens dataset added!**
> > >
> > > Dear reviewer,
> > >
> > > Thank you for your reply. We understand the need to add benchmark results on famous datasets with real missing data. Therefore we followed your recommendation (and thank you for the guidance) and added experimental results using the MovieLens 1m dataset.
> > >
> > > Note that the $k$NN$\times$KDE is only designed to handle numerical values as the name of this manuscript suggests. Therefore, we came up with an ad hoc solution to work with the data provided by the MovieLens dataset which are almost exclusively categorical features. Also, we only considered users for the rows of our matrix in order to look for neighbours, hence disregarding information on the similarity between movies. For the detail of the methodology and the results, please refer to the newly added Appendix F of our manuscript, as well as the Jupyter Notebook added into the Supplementary Materials. We used the leave-one-out scheme to evaluate the performances of the $k$NN$\times$KDE on the MovieLens dataset and obtained an average RMSE of 0.975.
> > >
> > > We hope that this addresses your point.
> > > Sincerely yours,

---

### Decision · Action_Editors · 2023-06-12

**Recommendation:** Accept with minor revision

**Comment:**

The method is supported by exhaustive experiments: "The authors have conducted extensive numerical experiments on synthetic toy data sets and heterogeneous data sets" [VbMm]; "Extensive empirical comparisons with other established imputation methods investigate the performance of the proposed approach under a variety of missingness patterns and conditions" [RUkZ].

Despite its simplicity, the performances of kNN*KDE are very good: "the superiority of KNNKDE over existing numerical imputation methods in imputing missing data with multi-modal distributions" [VbMm]; "The approach proposed is appealing in its simplicity", "The empirical evaluation is very convincing" [RUkZ]; "the proposed method is simple and easy to understand", "demonstrates its effectiveness in various simulated datasets and real datasets with simulated missingness" [PGgQ].

The reviewers praised the clarity of the paper: "The paper is well-written, logically structured, and easy to follow." [VbMm]; "I found the paper easy to follow" [PGgQ].

The following important points were raised by the Reviewers and led to discussion:
- no experiments with categorical variables [RUkZ]: the authors acknowledge that kNN*KDE cannot handle categorical variables yet, which is a limitation of the method.
- calibration of the hyperparameters [VbMm]: the authors clarified this point: "the authors make additional explanations for the mild weakness in the original version of the paper" [VbMm].
- lack to comparison with matrix completion method [PGgQ]: some experiments were included by the authors.

The code is available online and can easily be used by practitioners. Its performance is good. Thus, I believe the method is of certain practical interest, and I support the publication of the paper in TMLR. This opinion is supported by [VbMm] and [PGgQ]. However, [RUkZ] still believe that some problems should be fixed before publication. I will thus opt for "minor revision", so that the authors can address these problems.

I copy and paste the comments of [RUkZ] in his/her recommendation. Please address them as best as you can before sending the revised version.

>> The synthetic data experiments suggest that the baseline methods are generally not able to model the data sets with non-linear dependency structure, while the proposed method is (cf figures 2 and 3).

>> I think I understand the arguments of the authors but I still feel that the argument and illustration should be made such that readers do not get the impression that the nonlinear baselines (kNN, forests) cannot model these manifolds while the proposed method can. This is not true and presenting synthetic experiments in a way that suggest that was the reason for me to give the „leaning to reject“ vote.

>> On a different note, the authors emphasized in the discussions and the manuscript that their method is superior as it allows to model the posterior distribution of to-be-imputed values - this is also true for some other baselines that perform poorly in these experiments. Also here the reason for that worse performance could have been explained better. For instance the MICE implementation uses a linear model, which explains its poor performance.

>> So to summarize the comparisons with the baselines in real-data experiments suggest that the proposed method is not really much better than existing well established, simple and more scalable methods. And the synthetic experiments are at least a bit misleading: Some advantages of the proposed method are shared with some baseline methods (and do not help) and some differences in performance are suggesting that baseline methods are worse than they could be. The authors acknowledged these points but did not change the relevant figures 2 and 3.

**Audience:**

Very large audience. Missing data are one of the most recurrent practical problem in statistics and ML. The paper is very clear and should be of interest to applied statisticians and machine learners.

**Claims And Evidence:**

The authors propose a new method for missing data imputation. It's based on kNN combined with kernel density estimation (KDE). The strength of an approach based on density estimation is that it is able to handle multimodal distribution and complex dependencies between the variables. The method is tested on synthetic and real data, with convincing results.